# The KMT2F histone methyltransferase interacts with the RNA polymerase I machinery to promote ribosomal RNA transcription

Kaisar Ahmad Lone[1,2], Amit Mahendra Karole[1¤a], Geethanjali Ravindran[1¤b], Shweta Tyagi[1]*

**1** Laboratory of Cell Cycle Regulation, BRIC-Centre for DNA Fingerprinting and Diagnostics (CDFD), Uppal, Hyderabad, India, **2** Graduate Studies, Regional Centre for Biotechnology, Faridabad, India

¤a Current Address: Aragen Life Sciences Limited, Hyderabad, India
¤b Current Address: Department of Psychiatry, University of Illinois Chicago, Chicago, Illinois, United States of America
* shweta@cdfd.org.in

## Abstract

Trimethylation of histone 3 lysine 4 (H3K4me3) is a mark of active transcription, and its regulatory role in RNA polymerase II-mediated transcription has been well-studied. However, if and how this mark regulates RNA polymerase I (RNA Pol I) is not known. Here, we used customized genome assemblies for rDNA to demonstrate that KMT2A and KMT2F bind to entire rDNA loci. The binding of these enzymes was mirrored by the binding of H3K4me2 and H3K4me3 marks. Using biochemical assays, we demonstrate the interaction of KMT2-specific subunits with RNA Pol I transcriptional machinery. Our findings reveal KMT2F as the primary KMT depositing the H3K4me3 on rDNA. Loss of H3K4me3 adversely affects the epigenetic landscape and leads to repression of the rDNA locus. Mechanistically, using mammalian cells as a model system, we demonstrate that KMT2F promotes the formation of the pre-initiation complex by RNA Pol I. Our findings highlight the thus far undiscovered role of H3K4me3 in the transcriptional initiation of rDNA genes.

## Introduction

Within the cell, the intricate coordination of cellular growth and proliferation occurs through a precisely regulated series of steps called ribosome biogenesis. Deregulation within this elaborate process is associated with a diverse spectrum of human diseases, including cancer [1–5]. The control of ribosomal RNA (rRNA) transcription plays a central role in the complex pathway of ribosome biogenesis. The initial and crucial regulatory stage in ribosome biogenesis occurs through the transcription of rRNA genes (rDNA) by RNA polymerase I (RNA Pol I) within the dynamic membrane-less structure called the nucleolus [4,6]. The average diploid human genome harbors approximately 400 tandemly arranged repeats of rRNA genes,

**Data availability statement:** All relevant data are within the paper and its Supporting information files. The ChIP-seq data for KMT2A/F are deposited in the NCBI GEO database (GSE261933). Other data sets used for ChIP-seq are accessed from the following databases: SRP004987/SRA027342 (Pol I & UBF), GSE198645 (H3K4me2/me3), GSE186758 (WDR82), GSE60897 (WDR5), GSE149484 (H3K79me2), DRA004872 (DOT1L), while the RNA-seq data is utilised from GSE231942.

**Funding:** This work was supported by the DBT/ Wellcome Trust India Alliance Senior Fellowship to S.T. [IA/S/18/2/503981 http://www.indiaalliance.org/] and BRIC-CDFD core funds. CSIR SRF Award to K.A.L [CSIR Ref No.09/724(0141)/2019-EMR-I Dtd.24/12/2019]. The funders had no role in study design, data collection and analysis, decision to publish, or preparation of the manuscript.

**Competing interests:** The authors have declared that no competing interests exist.

**Abbreviations:** ChIP, chromatin immuno-precipitation; GO, Gene Ontology; GOA, Gene Ontology Annotation; GST, glutathione S-transferase; hTERT, human Telomerase everse Transcriptase; H3K4me3, trimethylation of histone 3 lysine 4; IFS, immunofluorescence staining; IGS, intergenic spacer; KMT, lysine methyltransferase; qRT-PCR, quantitative real-time PCR; rDNA, ribosomal DNA; RNA Pol I, RNA polymerase I; RNA Pol II, RNA polymerase II; rRNA, ribosomal RNA; SD, standard deviation; SL1, Selectivity Factor 1; TAD, transcriptional activation domain; TAFs, transcription-associated factors; TBP, TATA binding protein; UBF, upstream binding factor; UTR, untranslated region.

distributed across five acrocentric chromosomes [6]. Each Mammalian transcription unit (43 kb), consists of a coding region spanning 13.3 kb interspaced by 30 kb intergenic spacer (IGS). The coding region is responsible for encoding the 47S pre-ribosomal RNA (pre-rRNA), which undergoes subsequent processing to yield the mature 5.8S, 18S, and 28S rRNA. The coding unit is flanked by the non-coding regulatory elements, promoter, and a terminator sequence in the IGS. In addition to the human ribosomal DNA (rDNA) core or 47S promoter, the rDNA promoter contains a spacer promoter, which is separated by enhancer repeats and harbors essential signal sequences for transcription initiation and termination [7–12]. The main rRNA promoter (referred to as 47S promoter from hereon) directs the pre- rRNA synthesis, while the role of the spacer promoter is not clear.

The activity of RNA polymerase I is intricately regulated through interactions with numerous auxiliary factors. These factors play a crucial role in promoter recognition and contribute significantly to the processes of transcription initiation, elongation, and termination [13,14]. One of the primary regulators of rDNA transcription is the upstream binding factor (UBF). It plays an important role in various stages of this process, including pre-initiation complex (PIC) assembly, promoter escape, and elongation [15,16]. Additional elements within the RNA polymerase I machinery, such as the Selectivity Factor 1 (SL1) complex and RNA polymerase I-specific transcription initiation factor RRN3, are crucial for facilitating the initiation of RNA polymerase I. The 300 kd SL1 complex, composed of the TATA binding protein (TBP) and four transcription-associated factors (TAFs) TAF1A/TAF$_I$48, TAF1B/TAF$_I$63, TAF1C/TAF$_I$110 and TAF1D/TAF$_I$41, serves as the foundation for the creation of a proficient initiator complex on the promoter [17–20]. RRN3, in turn, facilitates the recruitment of RNA Pol I to the promoter and serves as a bridge between RNA Pol I and the SL1 complex anchored at the promoter. Therefore, the SL1-RNA Pol I-RRN3 ternary complex anchors at the promoter bound by UBF, leading to the activation of rDNA transcription [21].

RNA Pol I mediates its transcriptional activity at the rDNA locus by repeatedly engaging on tandem repeats of rDNA. Significantly, not all repeats exhibit transcriptional activity; nearly half of them are maintained in a transcriptionally poised state, primarily through epigenetic mechanisms like histone (H3K9me, H4K20me) or DNA methylations [22]. The rDNA that is actively transcribed is typically linked with euchromatin, exhibiting hypomethylation of rDNA and carrying histone modifications commonly associated with gene activation, such as H3K4 methylation and H3K9 acetylation [22]. Examination of rDNA chromatin structure through genomic analysis has unveiled the existence of H3K4 methylation marks at both rDNA promoters and intergenic regions [7]. Nevertheless, the specific role played by H3K4 methylation in the activation of rDNA transcription remains unclear. The existence of H3K4 methylation marks specifically at both promoters and intergenic regions of rDNA repeats suggests that the H3K4 lysine methyltransferase (KMT) complex could have a significant role in regulating rDNA transcription. But how this regulation is brought about by the H3K4 KMTs is not known.

H3K4 methylation is mediated by the COMPASS/Mixed Lineage Leukemia (MLL)/ lysine methyltransferase 2 (KMT2) family of proteins. The mammalian COMPASS

family consists of six members MLL1/KMT2A, MLL2/KMT2B, MLL3/KMT2C, MLL4/KMT2D, SET1A/KMT2F and, SET1B/KMT2G each actively engaged in complexes that include four common subunits collectively referred to as WRAD (WDR5, RBBP5, ASH2L, and DPY30) [23–25]. These family members modulate transcription primarily via their Su(var)3–9, Enhancer-of-Zeste, Trithorax (SET) domain, though some members like KMT2A and KMT2B also possess a transcriptional activation domain (TAD). Interestingly, KMT2 family members exhibit variations in their methyltransferase activity. KMT2F is the global KMT, mainly responsible for depositing trimethylation of histone 3 lysine 4 (H3K4me3) marks on promoters throughout the genome. KMT2A and KMT2B, on the other hand, demonstrate selective H3K4me3 activity at specific loci, such as Hox loci [26–28]. Several reports suggest that H3K4 methylation deposited by the members of the COMPASS family plays an important role in facilitating the assembly of the transcription pre-initiation complex and the recruitment of RNA polymerase II (RNA Pol II) to specific gene promoters [29,30]. Recent study depleted shared subunits within the COMPASS family to show that the abrupt decrease in H3K4me3 shows no discernible effects on the transcriptional initiation of RNA Pol II; nevertheless, it leads to a widespread decrease in transcriptional output, an increase in RNA Pol II pausing, and a slowdown in elongation, underscoring the crucial role of H3K4 methylation, mediated by the KMT2 family, in modulating RNA Pol II activity [31,32]. While there is increasing evidence supporting the role of H3K4 methylation in regulating RNA Pol II activity, the role of H3K4 methylation in connection with RNA Pol I regulation remains largely unknown.

Here, we undertook studies to understand how RNA Pol I transcription is regulated by the H3K4 KMT enzymes. We made use of customized genome assemblies for rDNA to demonstrate that KMT2A and KMT2F bind to the entire rDNA loci [33]. The binding of these enzymes correlated closely with the H3K4me2 and H3K4me3 marks. In order to demonstrate direct association, we used biochemical assays to examine the interaction between the common subunit, WDR5, as well as KMT2A and KMT2F, with the members of RNA Pol I machinery. Our transcriptional assays revealed a reduction in rRNA levels upon depletion of KMT2 members. Through mutational analysis, we discovered that KMT2F functions through its SET domain to regulate rDNA transcription. Our knockdown studies substantiated that KMT2F is the primary enzyme responsible for regulating the deposition of H3K4me3 on rDNA. Subsequently, we investigated the impact of perturbing the epigenetic landscape of rDNA and the formation of the RNA Pol I PIC, using ChIP assays. Our findings indicate that methyltransferase activity of KMT2F is requisite for PIC formation and ensuing transcription by RNA Pol I. Thus, our results reveal H3K4me3 mark as a key determinant in RNA Pol I PIC formation and active transcription from this locus.

## Results

### KMT2A and KMT2F bind to the ribosomal DNA (rDNA) locus

The nucleolus is a distinct structure and site of transcription and processing of rRNA. Proteins involved in rRNA transcription are often enriched at the nucleolus. We and others have previously reported that KMT2A localized to the nucleolus in human cells [34–36]. We undertook further experiments here to interrogate if KMT2A has a functional role in rRNA transcription. As KMT2F is the global H3K4 KMT, we included it in our analysis here. Using immunofluorescence staining (IFS), we observed that KMT2A and KMT2F co-localized with nucleolar marker protein B23 in U-2OS cells (Fig 1A). Specific siRNAs directed against KMT2A or KMT2F transcripts reduced nucleolar localization of both proteins, confirming the specificity of this localization in U-2OS and HEK293 cells (S1A–S1D Fig). We further performed IFS on various other cell lines, including MCF-7, HeLa, and non-transformed IMR90-tert, and observed that both proteins localized to the nucleolus in these cells (S1E and S1F Fig). To further validate nucleolar localization, we expressed GFP-tagged KMT2A or KMT2F together with dsRed-tagged B23. Both KMT2A and KMT2F showed clear colocalization with B23 (S1G Fig). Collectively, the analyses across multiple cell lines and exogenous expression demonstrate consistent nucleolar localization of KMT2A and KMT2F.

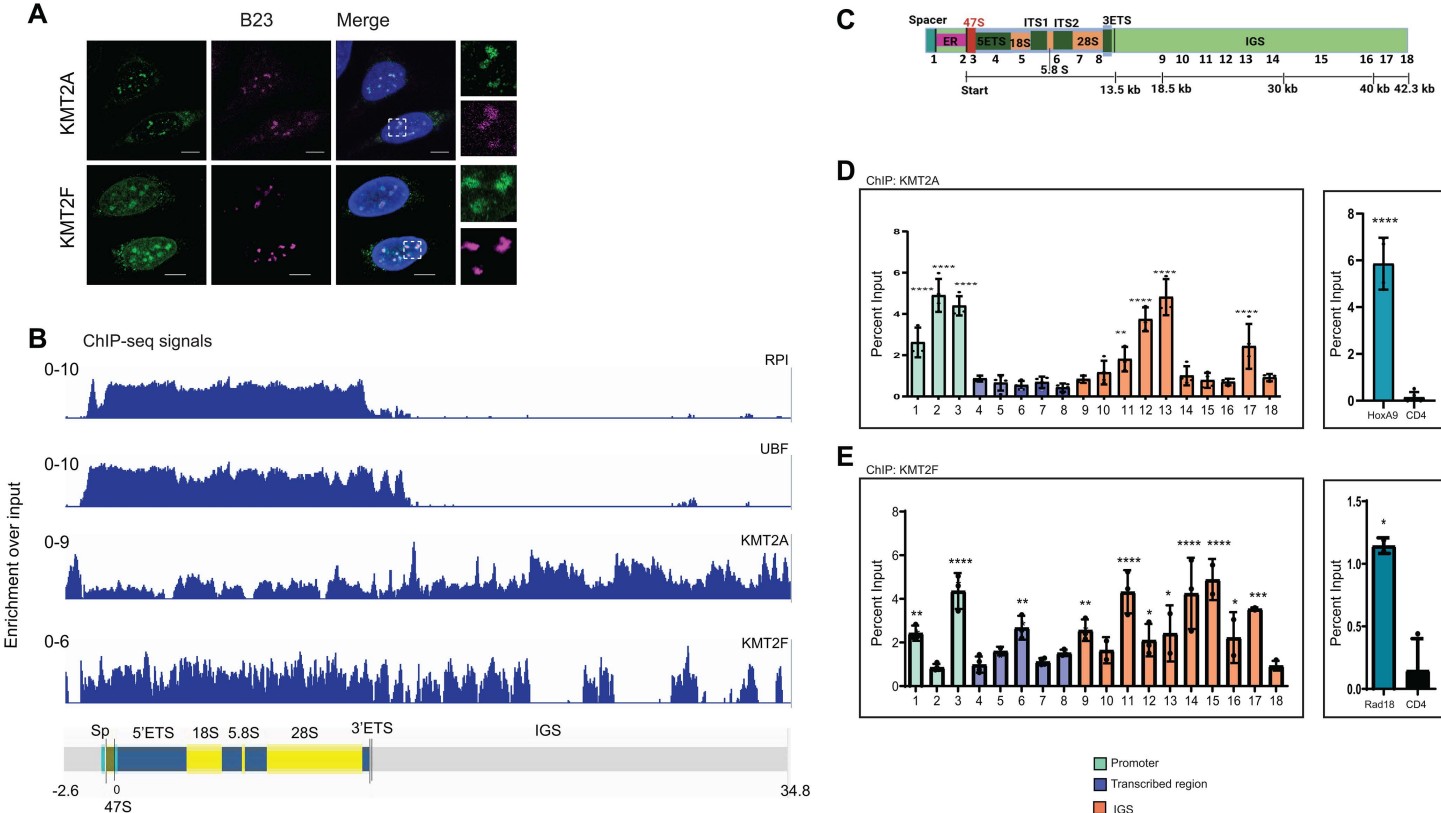

**Fig 1. KMT2A and KMT2F bind to the human ribosomal DNA (rDNA) repeats. A.** Immunofluorescence staining (IFS) of endogenous KMT2A and KMT2F is shown. U-2OS cells were co-stained with either KMT2A or KMT2F and nucleolar marker B23. DNA was counterstained with 4,6-diamidino-2-phenylindole (DAPI, blue). Scale bar, 5 µm. The highlighted box is shown as a zoomed-in image on the right. **B.** Chromatin immunoprecipitation-sequencing (ChIP-seq) maps, illustrating the binding of RNA Pol I (RPA116), UBF and KMT2's at the human rDNA locus, are shown. ChIP enrichment for each factor is expressed as the ratio of immunoprecipitated ChIP-seq signal to Input DNA ChIP-seq signal. The vertical scale in each panel indicates the enrichment of each component relative to the Input DNA dataset. On the X axis, tracks were truncated at 2.6 kb upstream of the main promoter (47S), denoted as −2.6 (kb), and at 34.8 (kb) downstream (also see Materials and methods and S2 Fig). The underlying ChIP-seq data can be found under Accession numbers SRP004987/ SRA027342 and GSE261933. **C.** Schematics representing 43kb human rDNA repeat drawn to scale. Each human rDNA repeat unit contains a 13kb sequence encoding the precursor ribosomal RNA, and Intergenic spacer region (IGS) spanning 30 kb. Primers (1 to 18) used for qRT-PCR represent different regions of the rDNA repeat. Co-ordinates and sequence of each primer are given in S1 Table. Spacer: spacer promoter; ER: Enhancer repeat; 47S: core promoter; ETS: External transcribed spacer; ITS: Internal transcribed spacer. **D, E.** ChIP analysis of KMT2A (**D**) and KMT2F(**E**) is shown. Primers 1-18 were used to assess the binding of KMT2A/KMT2F over the 43kb human rDNA repeat. Spacer promoter, terminator element (T0) and 47S (core) promoter are represented by primers 1–3 (green), respectively; transcribed region: 4–8 (blue) and IGS by 9–18 (orange). *HOXA9* and *RAD18* were used as positive control regions for KMT2A and KMT2F binding respectively. *CD4* was used as negative control for KMT2A and KMT2F ChIP. Data is presented from three or more experiments. Significance was calculated for each primer with respect to the negative control CD4 primer using one-way ANOVA with Dunnett's multiple comparison test. Error bars represent SD. *$P \leq 0.05$, ** $P \leq 0.005$, ***$P \leq 0.0005$, ****$P \leq 0.00005$. The underlying data pertaining to D and E can be found in S1 Data.

As rDNA is repetitive in nature, it has been left unannotated in most genome assemblies. However, the rDNA repeat is a large locus to be mapped with chromatin immunoprecipitation (ChIP) alone. In order to interrogate whether the KMT2 proteins bind to rDNA repeats or not, we performed paired-end ChIP followed by sequencing (ChIP-seq) for KMT2A and KMT2F. The reads obtained were mapped to custom human genome assembly called hg38-rDNA (hg38) in which a single unit of rDNA has been added (Fig 1B and 1C) [33]. We included the ChIP-seq analyses of RNA Pol I second largest sub-unit, RPA 116 (referred to as RPI in Fig 1B) and UBF as a positive control (Fig 1B). Both RNA Pol I-and UBF-bound reads mapped to the transcribing unit of rDNA as reported previously (Figs 1B and S2) [7,37]. We observed that KMT2A and

PLOS Biology

KMT2F exhibited binding to the entire region of rDNA sequence, including the RNA Pol I promoter, transcribed unit and the IGS (Figs 1B and S2). KMT2F showed a consistent binding in the RNA Pol I transcribed unit, with distinctive peaks in the IGS, whereas KMT2A bound throughout the RNA Pol I transcribed region but displayed higher peaks with continuous binding in the IGS region. To strengthen our confidence in the binding of these proteins to the rDNA locus, we utilized data from other KMT2 family-associated proteins. The WRAD complex plays a critical role in determining the binding specificity of KMT2 family members on chromatin [38]. For this we took advantage of ChIP-seq data for WDR5, a common component shared by all KMT2 family members [39]. Upon analyzing the ChIP-seq tracks, we observed significant WDR5 binding in the transcribed unit of the rDNA, along with a distinct peak in the IGS region (S2 Fig). Since KMT2F lacks direct DNA-binding domains, it is recruited to chromatin via the WDR82 protein [40]. WDR82, which uniquely binds to the KMT2F complex, was further analyzed using ChIP-seq data [41] to explore its binding pattern across the entire rDNA locus. We found that WDR82 showed higher enrichment in the transcribed unit (S2 Fig), indicating its possible role at the rDNA locus. Thus, our ChIP-seq data reveal subtle differences in the binding of KMT2A and KMT2F at the rDNA locus. To further validate the specificity of KMT2 family binding at rDNA, we used DOT1L as a control. Analysis of DOT1L ChIP-seq data revealed negligible occupancy of DOT1L across rDNA, thereby reinforcing the specificity of our rDNA binding analyses (S2 Fig) [42].

The above results gave us confidence that the KMT2 complexes bind to rDNA. However, as rDNA genome assemblies are still evolving, we validated our findings by performing ChIP analyses using select primers spanning the entire rDNA unit (Fig 1C and S1 Table). Primer #1 corresponded to the spacer promoter, #3 to the core 47S promoter, #4–7 to the gene body, #8 to termination repeats, and #9–18 spanned the IGS as shown (Fig 1C and S1 Table). We also used canonical targets as positive control (HoxA9 for KMT2A and Rad18 for KMT2F) and CD4 as a negative control. We first performed ChIP with RNA Pol I and UBF (S3A and S3B Fig). As expected, both proteins showed highly enriched binding on the rDNA promoter and transcribed region. This binding decreased abruptly after the termination signal (primer #8; S3A and S3B Fig). The results obtained here indicated that we are able to ChIP for proteins bound to rDNA successfully. We observed that in directed ChIP, KMT2A was significantly enriched at the RNA Pol I promoter and IGS (Fig 1D), whereas KMT2F bound primarily in the region of RNA Pol I promoter, and transcribed unit as well as IGS (Fig 1E). In order to assess if the role of these proteins is cell-type specific, we performed ChIP in non-transformed IMR90-tert cells, fibroblasts isolated from normal lung tissue and immortalized using human Telomerase Reverse Transcriptase (hTERT). Similar to our observations made in HEK293 cells (Fig 1D and 1E), KMT2A displayed binding primarily at the RNA Pol I promoter and IGS, whereas KMT2F bound throughout the rDNA repeat (S3C and S3D Fig). Taken together, our data indicate that KMT2A and KMT2F bind to the rDNA repeat locus, including the RNA Pol I promoter.

The ChIP-seq assembly spans the entire rDNA repeat, while our primers are chosen at select regions as described above (Fig 1C). To evaluate whether our ChIP-qPCR agrees with our ChIP-seq data, we performed a quantitative analysis of the ChIP-seq signal corresponding to each primer used for qPCR and compared these values with the ChIP–qPCR data (S4A and S4B Fig). Overall, the two datasets for KMT2A and KMT2F are broadly concordant, although some discrepancies are observed. Here, we would like to note that analysis of protein occupancy at the rDNA locus is technically challenging and associated with inherent limitations due to its highly repetitive nature. In the present study, ChIP-seq signals were aggregated across ~400 rDNA repeats and presented as collated data. Hence, minor differences between datasets likely reflect technical variability inherent to rDNA ChIP-seq analyses rather than true biological differences.

## rDNA locus is enriched for H3K4me2 and H3K4me3 mark

To understand if the binding of KMT2 proteins has a functional role on the rDNA locus, we performed similar analyses on previously published paired-end ChIP-seq data sets for H3K4me2 and H3K4me3 [43] using hg38-rDNA (hg38) genome assembly. We observed that H3K4me2 and H3K4me3 marks were present on the RNA Pol I promoter and transcribed region. This binding continued in the IGS region, eventually getting confined to distinct peaks (Figs 2A and S2). We further

PLOS Biology

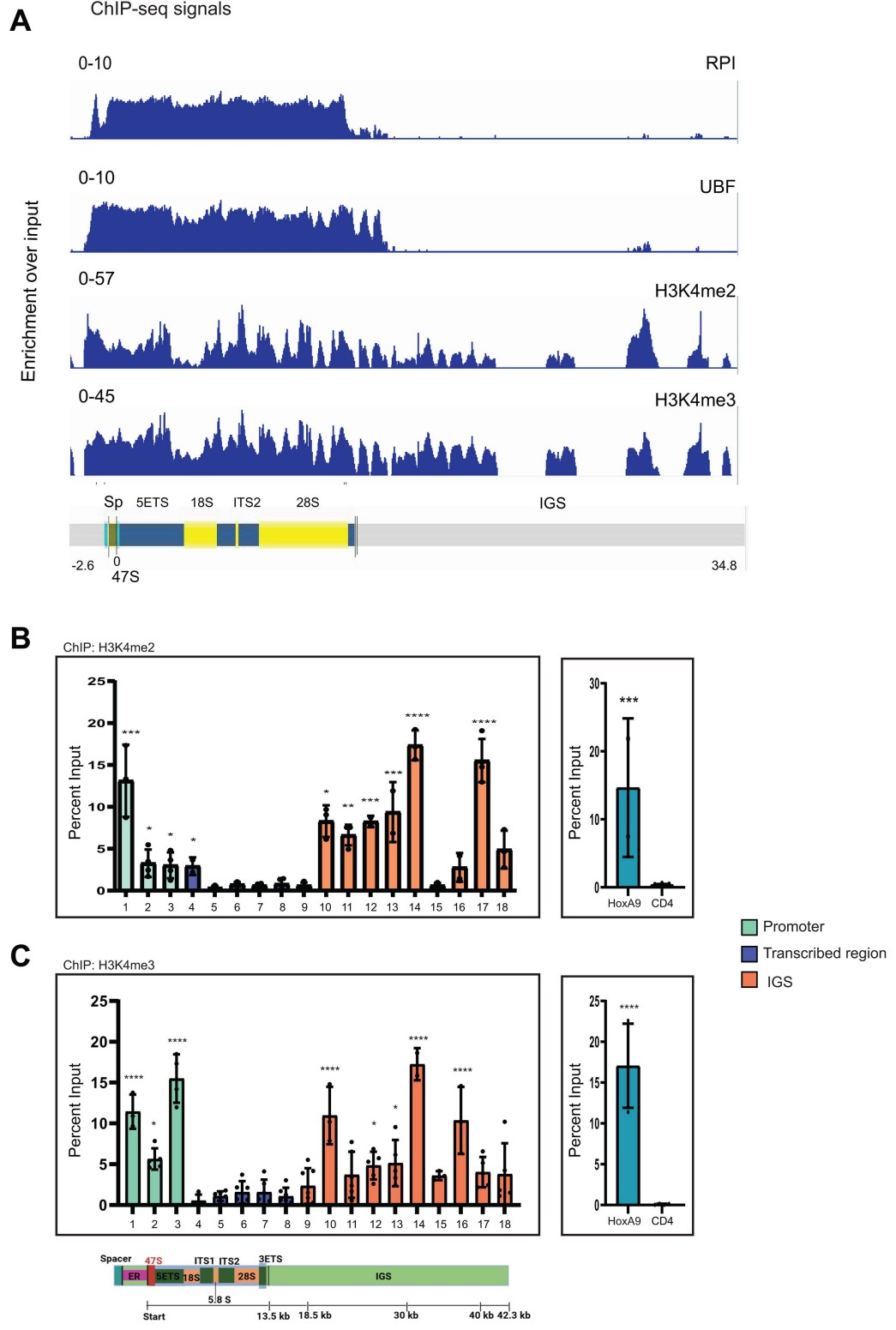

**Fig 2. Active Epigenetic signature of human rDNA. A.** ChIP-seq maps illustrating the binding of RPA116, UBF and distribution of active epigenetic marks (H3K4me2 and H3K4me3) at the human rDNA locus are shown. ChIP enrichment for each factor is analyzed as in Fig 1B. *Please note that*

*RPA116 and UBF tracks are duplicated from Fig 1B and shown here for ease of comparison.* (also see S2 Fig). The underlying ChIP-seq data can be found under Accession numbers SRP004987/ SRA027342 and GSE198645. **B, C**. Enrichment of H3K4me2 (B) and H3K4me3 (C) across human rDNA as revealed by ChIP followed by qRT-PCR. Primers labeled from 1 to 18 were used to access distribution pattern of H3K4me2 and H3K4me3 across 43 kb repeat. *HOXA9* and *CD4* were used as positive and negative control regions, respectively. Data shown as means ± SD from three or more than three individual biological replicates. Error bars represent SD. Significance was calculated for each primer with respect to the negative control CD4 primer using one-way ANOVA with Dunnett's multiple comparison test. *$P \le 0.05$, ** $P \le 0.005$, ***$P \le 0.0005$, ****$P \le 0.00005$. Green: RNA pol I promoter; blue: RNA pol I transcribed region; orange: IGS. The underlying data for B and C can be found in S1 Data.

used H3K79me2 ChIP-seq analysis as a negative control for histone modification enrichment, as H3K79me2 deposited by DOT1L is not known to be associated with rDNA chromatin. As expected, no enrichment of H3K79me2 was observed at the rDNA (S2 Fig).

Again, we validated our above findings for H3K4me2 and H3K4me3 marks by ChIP analyses and observed consistently high enrichment of these marks on rDNA promoter and IGS region in HEK293 (Fig 2B and 2C). Consistent with the presence of these marks, H3 could be detected in the entire rDNA repeat region (S3E Fig). To further correlate the ChIP-seq data with ChIP-qPCR for these two modifications, we quantified our ChIP-seq data for all the rDNA primers used in this study (discussed in the Materials and methods). We observed good concordance between our ChIP-seq and ChIP-qPCR data for H3K4me2/me3 marks, demonstrating that the ChIP-seq results are consistent with the ChIP-qPCR findings (S4C and S4D Fig).

## Comparative analysis of KMT2A, KMT2F, and H3K4 methylation at rDNA versus genome-wide regions

To determine whether KMT2 family histone methyltransferases regulate rDNA chromatin differently from canonical Pol II transcribed regions, we quantitatively analyzed ChIP-seq signals of KMT2A, KMT2F, H3K4me2, and H3K4me3 across rDNA and compared them with their distribution to the rest of the genome at Pol II promoters, gene bodies, and intergenic regions. In order to start this analysis, we divided the genome into different regions such as promoters, gene body region, intergenic regions, and the rDNA (discussed in the Materials and methods). As previous attempts to quantitatively measure the binding of the PIC proteins on the rDNA have been done for the RNA Pol I and UBF [7], we first did the quantitative analysis of Pol I and UBF. Pol I, exclusively restricted to the nucleolus, showed 99.5 per cent enrichment on the rDNA as compared to the rest of the genome (S5A Fig). Some fraction of UBF has been reported to be nucleoplasmic [7]. In our analysis, we observed that 90 per cent of UBF distribution on the rDNA, while it showed 10 per cent nucleoplasmic binding, comprising binding on the promoters, gene body and intergenic regions (S5B Fig). Quantitative analysis of PIC component proteins gave us confidence to further analyze the binding of KMT2 proteins and associated modifications on the rDNA compared to the rest of the genome. Our analysis revealed that a substantial fraction of chromatin-associated signal for KMT2A, KMT2F, H3K4me2, and H3K4me3 localizes to rDNA compared with the rest of the genome. Specifically, 13% of total KMT2A, 22% of KMTF, 7% of H3K4me2, and 15% of H3K4me3 ChIP-seq signal was detected over rDNA (S5C–S5F Fig), despite rDNA representing a minute fraction of the genome. The remaining signal for each factor was distributed across Pol II promoters, gene bodies, and intergenic regions, consistent with their known genome-wide binding patterns [44]. This disproportionate enrichment underscores rDNA as a major chromatin target of KMT2 family methyltransferases and H3K4 methylation relative to conventional genomic compartments.

## H3K4 KMT complex interacts with RNA Pol I machinery

In order to establish a regulatory role of our KMTs in RNA Pol I-mediated transcription, we tested if these complexes specifically interact with RNA Pol I machinery. RNA Pol I holo- complex is composed of 14 subunits, out of which 7 are unique to RNA Pol I (hereafter called RNA Pol I-specific subunits; Fig 3A). These include RPA 194, RPA 116, RPA12, RPA49, RPA34, RPA43, and RPA14 (RPA 14 was discovered in yeast; however, it has not yet been characterized in mammals;

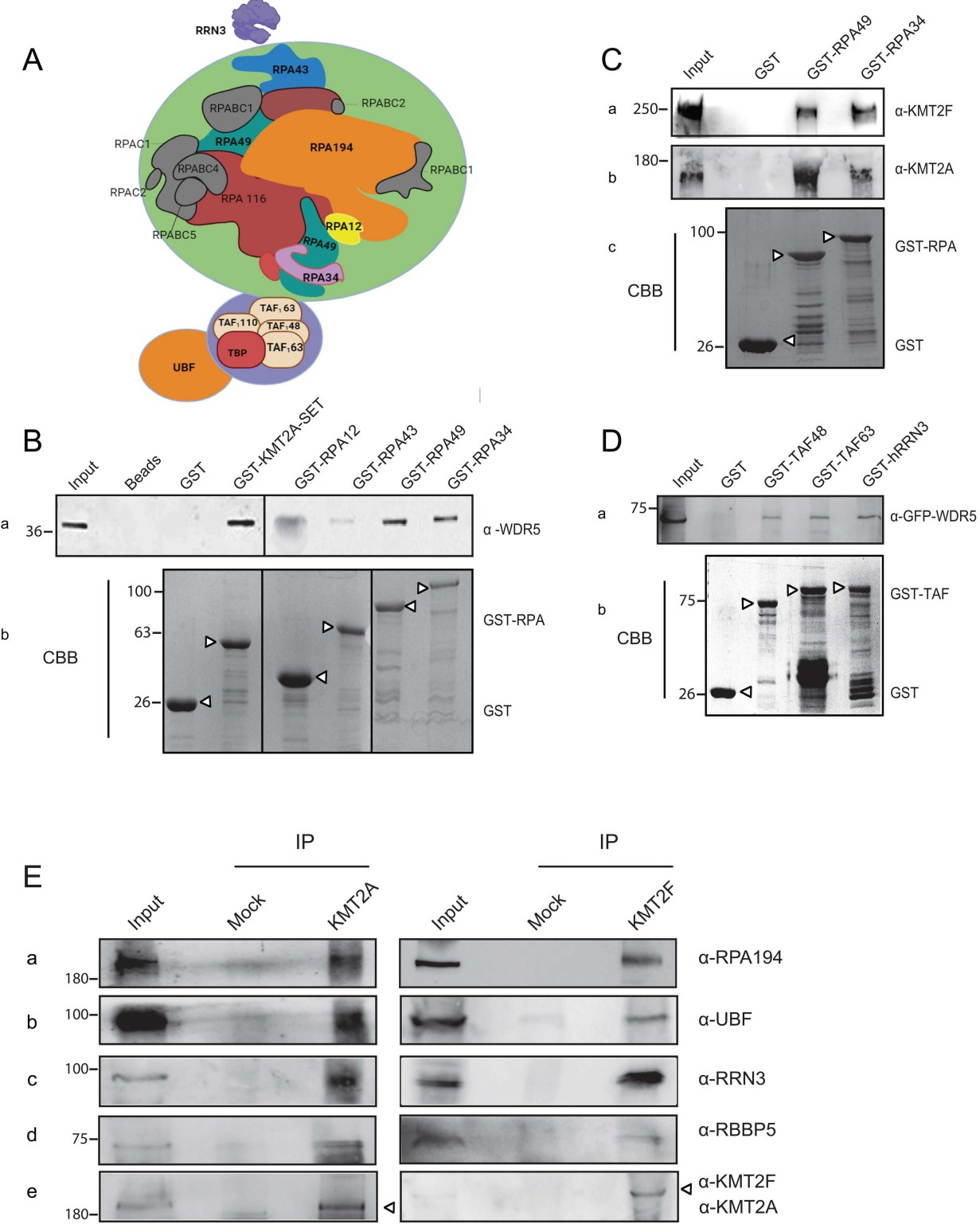

**Fig 3. RNA Pol I machinery interacts with KMTs. A.** Schematic showing the arrangement of RNA Pol I holo complex with its associated proteins. RNA Pol I has 14 subunits, out of which 7 are shared with Pol II and Pol III while the rest are unique to Pol I. Unique subunits are shown in color.

RPA190, RPA116 and RPA12 form the catalytic core of the RNA Pol I, whereas RPA34, RPA43 and RPA49 are classified as the peripheral subunits. UBF prepares the rDNA promoter for RNA Pol I docking. This is followed by binding of the SL1 complex, which consists of TATA binding protein (TBP) and four transcription-associated factors (TAFs) TAF1A/TAFI48, TAF1B/TAFI63, TAF1C/TAFI110 and TAF1D/TAFI41. hRRN3 identifies and recruits free RNA Pol I to the rRNA gene promoter that is already bound by UBF and SL1 complex. **B.** Bacterially expressed GST, GST-KMT2A-SET domain, GST-RPA12, GST-RPA43, GST-RPA49, and GST-RPA34 were incubated with the bacterially expressed and purified WDR5. The bead-bound proteins used for the pull-down are shown by Coomassie brilliant blue (CBB) staining (panel b) while interaction was visualized by western blot using anti-WDR5 antibody (panel **a**). *Please note that WDR5 and GST-RPA12 bands co-migrate and we believe that the smudge seen in GST-RPA-12 lane is produced by large amount of GST-RPA-12 and not WDR5*. **C.** Bacterially expressed GST and GST-tagged RPA49 and RPA34 were used to assess the interaction of KMT2A and KMT2F with RNA Pol I complex. This interaction was confirmed by immunoblotting with α-KMT2F and α-KMT2A antibodies shown in panel **(a, b)**. The amount of bead-bound protein used in the pull-down is shown by CBB staining in the bottom panel **c. D.** GST, GST-RRN3 and GST-tagged SL1 complex proteins (TAF48 and TAF63) were bacterially purified and incubated with whole cell extracts from cells ectopically expressing GFP-WDR5 shown in panel **(a)** (endogenous WDR5 could not be probed due to technical difficulties). The interaction was confirmed by immunoblotting with α-GFP. **(B–D)** The amount of protein taken for GST pull-down assay, stained with CBB, is shown as the bottom panel. Relevant protein bands are indicated with arrowheads. **E.** Nuclear extract prepared from HeLa cells was subjected to endogenous immunoprecipitation (IP) using antibodies against KMT2A and KMT2F. The anti- immunoglobulin G (IgG)- antibody was used as control (Mock). The immunoblot was probed with α-RPA194, α-UBF, α-RRN3, α-RBBP5, and α-KMT2A or α-KMT2F antibodies. RBBP5 was used as a positive control for anti-KMT2A and anti-KMT2F IP. **(B–E)** The molecular weight marker is indicated on the left. The uncropped blots from B to E are shown in the S1 Raw Images.

[45]. The remaining subunits are shared between all RNA polymerases. We tried to express all six RNA Pol I-specific subunits as N-terminal Glutathione S-transferase (GST) fusions in bacteria. However, technical challenges limited us to the successful expression of RPA12, RPA43, RPA49, and RPA34. Interestingly, three of these subunits are classified as the peripheral subunits of RNA Pol I and have been implicated in RNA Pol I initiation and elongation [46]. We tested in vitro interactions between bacterially expressed GST-tagged RNA Pol I-specific subunits with bacterially expressed WDR5 (Fig 3B). We used GST-KMT2A SET domain as a positive control and GST alone as a negative control. We observed that WDR5 specifically interacted with RPA49 and RPA34 (Fig 3B). As RPA49 and RPA34 showed consistent interaction with WDR5, we chose these subunits to pull down the SET-domain containing subunits of H3K4 KMT enzymes. Remarkably, GST-RPA49 and GST-RPA34 could pull-down the high molecular weight proteins KMT2A and KMT2F robustly, indicating that our proteins specifically interacted with the RNA Pol I subunits in the cells (Fig 3C).

In addition to the RNA Pol I enzyme, rDNA transcription requires UBF, SL1 complex, and transcription initiation factor hRRN3 [22]. SLI-RNA Pol I-hRRN3 ternary complex docks at UBF-bound RNA Pol I promoter that results in activation of transcription [19]. In order to establish that the KMTs interact with transcription-competent RNA Pol I complex, we checked for the interaction of TAF 48 and TAF 63 (subunits of SLI complex) and hRRN3 with WDR5. Consistent with our hypothesis, WDR5 interacted with SL1 complex and hRRN3 (Fig 3D). We could further prove that KMT2A and KMT2F interacted with various components of RNA Pol I by endogenous immunoprecipitation experiments where we pulled down KMT2A or KMT2F proteins using specific antibodies and verified their interaction with RPA194, UBF, and hRRN3 (Figs 3E and S6A). We examined the interaction of KMT2A and KMT2F with RNA polymerase II as a control for our endogenous co-immunoprecipitation experiments. Co-immunoprecipitation of KMT2A and KMT2F efficiently pulled down RNA polymerase II (Ser2-phosphorylated), and the extent of this interaction was comparable to the levels of RNA polymerase I recovered with these proteins (S6B Fig). Altogether, our results show that KMT2A and KMT2F interact with RNA Pol I and multiple components of its transcription machinery.

## KMT2F regulates rRNA transcription via its SET domain

We have shown so far that the H3K4 KMTs—KMT2A and KMT2F—bind to rDNA locus and interact with RNA Pol I transcription machinery. Given these results, we hypothesized that these KMTs activate transcription of rRNA. rRNA is transcribed into 13 kb long 47S precursor ribosomal RNA (pre-rRNA), which gets processed at its 5′ end, called 5′ external transcribed spacer (5′ETS) as soon as it gets transcribed to give rise to 45S pre-RNA [22]. Therefore, the 5′ETS can be used to detect the transcript level of pre-rRNA by quantitative real-time PCR (qRT-PCR).

We used RNAi to independently knock down KMT2A, KMT2F, and WDR5 in U-2OS cells. To check if other members of the KMT2 family had a role in transcriptional regulation of rRNA, we included KMT2B, and KMT2C in our experiments. The transcript and protein levels of KMT2 and WDR5 decreased by more than 50% (Figs 4A and S6C). As expected, we observed a decrease in the 5′ETS levels upon loss of KMT2A, KMT2F, and WDR5 (Fig 4B). Interestingly, loss of KMT2B, but not KMT2C, also resulted in a decrease in 45S rRNA levels, indicating that select members of the KMT2 family engage with RNA Pol I to regulate rRNA transcription.

Proceeding with our analysis and to exclude the possibility that regulation of rDNA transcription by KMT2 enzymes is cell-type-specific rather than universal, we examined rRNA synthesis in an independent cellular context. Depletion of KMT2A or KMT2F in HEK293 cells resulted in a significant reduction in 5′ETS-containing rRNA levels, indicating impaired rDNA transcription (S1C, S1D, S6D, and S6E Figs). These findings demonstrate that KMT2A and KMT2F-mediated regulation of rDNA transcription is conserved across cell types and is not restricted to a specific cellular system.

To investigate the specificity of the observed reduction of 5′ETS in KMT2A and KMT2F siRNA, we conducted complementation assays where we examined the rRNA transcripts in siRNA-treated cells that were expressing the respective recombinant full-length protein. Additionally, we utilized SET-domain deleted KMT2A/F protein(s) to gain insights into the role of the SET domain in rRNA transcription (S6F–S6I Fig). For this purpose, we used stable cell lines created in U-2OS cells [47,48]. To specifically target the endogenous KMT2A or KMT2F transcript with our siRNA treatment, and avoid affecting the recombinant transcript, we (i) designed siRNA to target the 3′ untranslated region (UTR) of the precursor mRNA for KMT2A; (ii) created recombinant siRNA-resistant KMT2F constructs by introducing silent mutations (as shown in S6H Fig).

We observed a substantial reduction in KMT2A and KMT2F transcript levels following treatment of U-2OS cells with KMT2A or KMT2F siRNAs, which was largely rescued in their respective stable cell lines (Figs 4C, 4D, S6G, and S6I). Corresponding to the reduction in KMT2A levels in U-2OS cells, as before, we observed a significant decrease in 5′ETS levels in the siRNA-treated U-2OS cells. This reduction was rescued in cells overexpressing full-length KMT2A and KMT2AΔSET but not in KMT2AΔTAD cells (Figs 4C and S6G), indicating that KMT2A regulates rDNA transcription through its TA domain. On the other hand, 5′ETS transcript levels were rescued in cells expressing full-length KMT2F, but not in cells expressing KMT2FΔSET, suggesting that KMT2F regulates rDNA transcription via the methyltransferase activity of its SET domain (Figs 4D and S6I).

To validate our results from the above experiments, we used an assay to measure ongoing rRNA transcription, where we gave a short pulse with [32P] orthophosphoric acid to label cells after RNAi treatment, extracted and ran RNA on agarose gel and visualized 47/45S pre-rRNA by autoradiography [49]. The assay enabled us to visualize freshly processed 32S, 28S, and 18S rRNA moieties though 45S rRNA was the predominant species seen (Fig 4E). We also used ethidium bromide gel to visualize 28S RNA as loading control, as it is largely made during previous rounds of transcription and remains stable over three days. Our results from pulse-chase studies concurred with our observations made with qRT-PCR (from Fig 4A–4D), and we observed a consistent decrease in the 45S rRNA upon knockdown of KMT2A (Figs 4E, 4F, 4H, S7B, and S7D) and KMT2F (Figs 4G, 4H, S7C, and S7D). This decrease was reproduced in HeLa, MCF-7, and IMR90-tert cell lines, indicating that KMT2A and KMT2F regulate 45S RNA irrespective of cell type (S7E and S7F Fig). We observed that while both KMT2A and KMT2F play a specific role in regulating rRNA transcription, only KMT2F did so in a SET-domain-dependent manner (Figs 4F, 4G, S7B, and S7C). In contrast, loss of TA but not SET domain in KMT2A resulted in decreased rRNA transcription (Figs 4F and S7B). Further, in agreement with our qRT-PCR results, we observed that depletion of KMT2B resulted in a decrease in 45S rRNA levels, but not that of KMT2C (Figs 4H and S7D). Consistent with these findings, we could detect KMT2B in the nucleolus (S7G Fig). Taken together, our findings indicate that only select members of the KMT2 family engage with RNA Pol I and that KMT2A and KMT2F regulate rRNA transcription through distinct mechanisms.

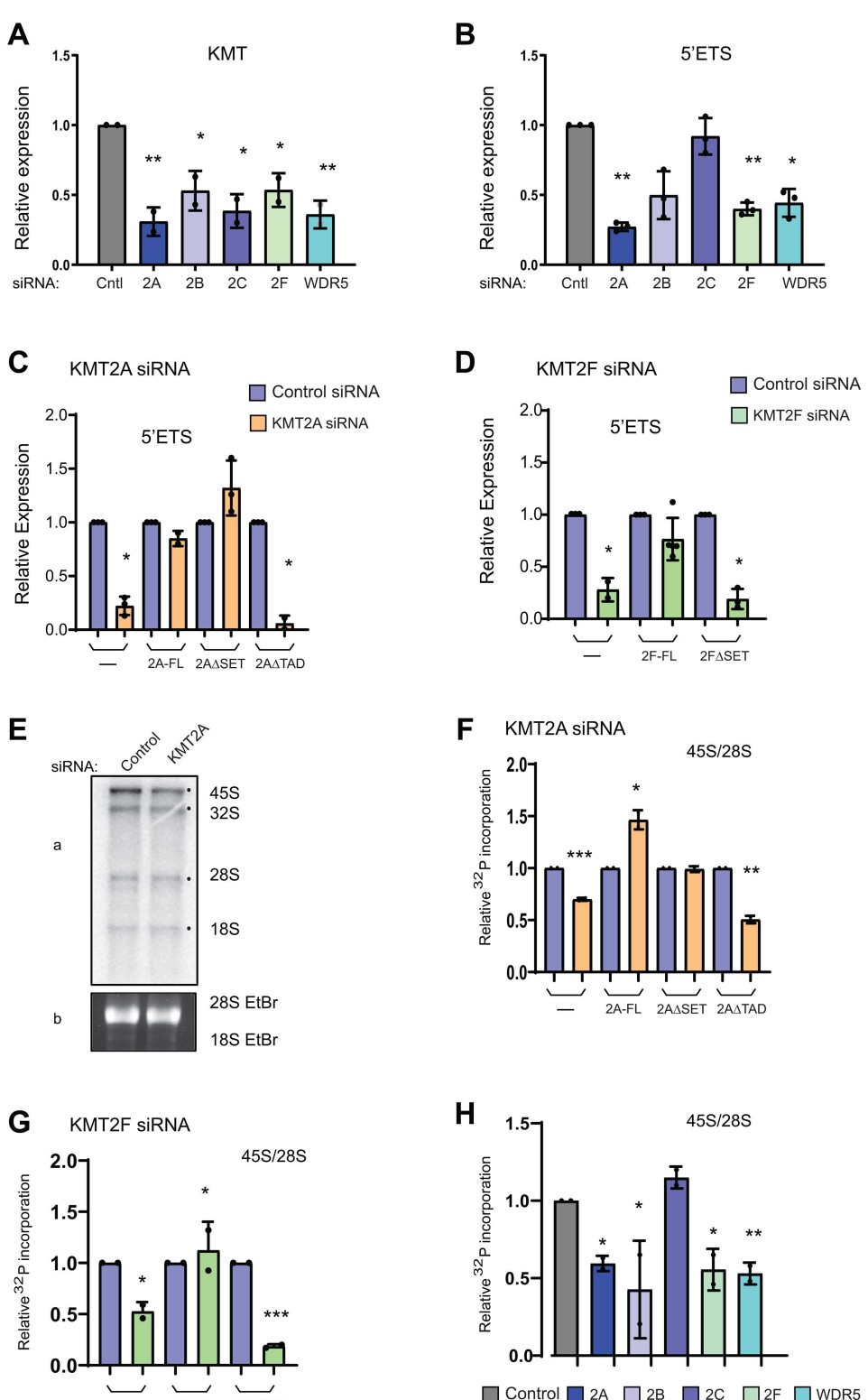

**Fig 4. RNAi-mediated depletion of KMTs affects ribosomal DNA transcription. A, B.** qRT-PCR analysis of KMTs and 5′ETS transcript levels after treating the cells with Control, KMT2A, KMT2B, KMT2C, KMT2F and WDR5 siRNA is shown. Error bars denote SD from three individual replicates.

Significance for each group was calculated with respect to the control using two-way ANOVA with Šídák multiple comparison test *$P \le 0.05$, ** $P \le 0.005$. **C, D.** qRT-PCR analysis of 5′ ETS transcript levels in various KMT2A and KMT2F mutant cell lines treated with specific siRNAs is shown. **C**. KMT2A depleted cells were complemented with stably expressing KMT2A full length (2A-FL), KMT2AΔSET (2AΔSET), and KMT2AΔTAD (2AΔTAD) cells and 5′ ETS transcript levels were analyzed as shown. **D**. Shows 5′ ETS transcript levels in control cells, KMT2F full length (2F-FL), and KMT2FΔSET (2FΔSET) cells. Error bars denote SD from three individual replicates. *$P \le 0.05$, (two-tailed Student $t$ test). **E.** Representative autoradiogram showing the $^{32}$P orthophosphate labeled ribosomal RNA upon loss of KMT2A. U-2OS cells were treated with Control and KMT2A siRNA. Extracted total RNA was run on agarose gel and visualized by autoradiography (panel **a**). Ethidium bromide stained 28S RNA levels were determined by running a formaldehyde agarose gel (panel **b**). The uncropped gels are shown in th S1 Raw Images. **F.** Effect of mutational analysis of KMT2A on rRNA transcription. Densitometry quantification of 32P incorporated 45S rRNA levels in endogenous KMT2A-depleted Control (U-2OS), KMT2A full length (2A-FL), KMT2AΔSET (2AΔSET) and KMT2AΔTAD (2AΔTAD) cells is shown. The relative intensity was measured by normalizing 45S levels with 28S rRNA loading control. Each value is an outcome of two independent experiments. SD is represented by error bars. The statistical significance was calculated with respect to the control by two-tailed Student $t$ test *$P \le 0.05$, **$P \le 0.005$, ***$P \le 0.0005$. **G.** Effect of mutational analysis of KMT2F on rRNA transcription. Shown is densitometry quantification of $^{32}$P incorporation into 45S rRNA levels in endogenous KMT2F-depleted U-2OS cells, which were complemented with stably expressing KMT2F full length (2F-FL), KMT2FΔSET (2FΔSET). The statistical significance was calculated with respect to the control by two-tailed Student $t$ test *$P \le 0.05$, ***$P \le 0.0005$. **H.** The 45S rRNA transcript levels upon loss of different KMT2 family members are shown. U-2OS cells were treated with KMT2A, KMT2B, KMT2C, KMT2F and WDR5 siRNAs. **(E–G)** $^{32}$P incorporation into 45S rRNA level was measured by densitometry. The relative intensity was measured by normalizing 45S levels with 28S rRNA loading control. Each value is an outcome of two independent experiments. SD is represented by error bars. The statistical significance was calculated with respect to the control by a two-tailed Student $t$ test *$P \le 0.05$, **$P \le 0.005$. The underlying values pertaining to A–D and F–H can be found in the S1 Data.

## KMT2A and KMT2F regulate rDNA independently of ribosome biogenesis pathway remodeling: Direct effect on the rDNA transcription rather than indirect effect

KMT2A and KMT2F regulate RNA Pol II transcription and hence, expression of a large number of genes [29,31,40]. To determine whether the effects of KMT2A and KMT2F on rDNA transcription arise indirectly through alterations in ribosome biogenesis programs, we analyzed our recently published RNA-seq datasets [48] following depletion of either KMT2A or KMT2F. Comparative transcriptomic profiling revealed no significant changes in the expression of genes associated with the regulation of ribosome biogenesis, rRNA processing, or ribosomal protein production between control and knockdown conditions (S5G and S5H Fig). Pathway enrichment analyses further confirmed the absence of differential regulation of ribosome biogenesis-related pathways. These findings indicate that loss of KMT2A or KMT2F does not globally perturb the transcriptional programs governing ribosome biogenesis. Taken together with their direct enrichment at rDNA and associated changes in 45S transcript, these results support a model in which KMT2A and KMT2F regulate rDNA transcription through direct chromatin-based mechanisms rather than indirect remodeling of the ribosome biogenesis machinery via RNA Pol II.

## KMT2F is the H3K4me3 methyltransferase for rDNA locus

Several reports indicate that rDNA bears the H3K4me3 marks and these marks are involved in shaping the epigenetic landscape of this loci [7,50,51]. Our data indicate that some members of the KMT2 family regulate rRNA transcription, and we have shown that KMT2F utilizes its SET domain, to regulate rRNA transcripts. Further, KMT2F bind to the rDNA and interacts with RNA Pol I. To investigate the role of this protein in detail, we used shRNA-mediated-knockdown for KMT2F as described previously [48]. Consistent with the reduced protein levels on the blots, the chromatin binding of KMT2F was drastically reduced on rDNA transcribed by RNA Pol I (Figs 5A, 5B, S8A). Remarkably, consistent with the role of the SET domain of KMT2F in our rRNA transcript analyses, knockdown of KMT2F resulted in a drastic reduction of H3K4me2 and H3K4me3 levels on the whole rDNA loci, although the global levels of these modifications remained unchanged (Figs 5C, 5D, and S8B–S8D). Thus, the loss of KMT2F resulted in the reduction of H3K4methylation levels throughout the RNA Pol I transcribed region when compared with the control (Fig 5C). Hence, our results indicate that KMT2F is the primary H3K4 methyltransferase of rDNA.

As our transcriptional analyses indicated that rDNA transcription occurs independently of KMT2C, we decided to use KMT2C as a negative control for epigenomic analyses at rDNA. To validate the specificity of our ChIP assays, we

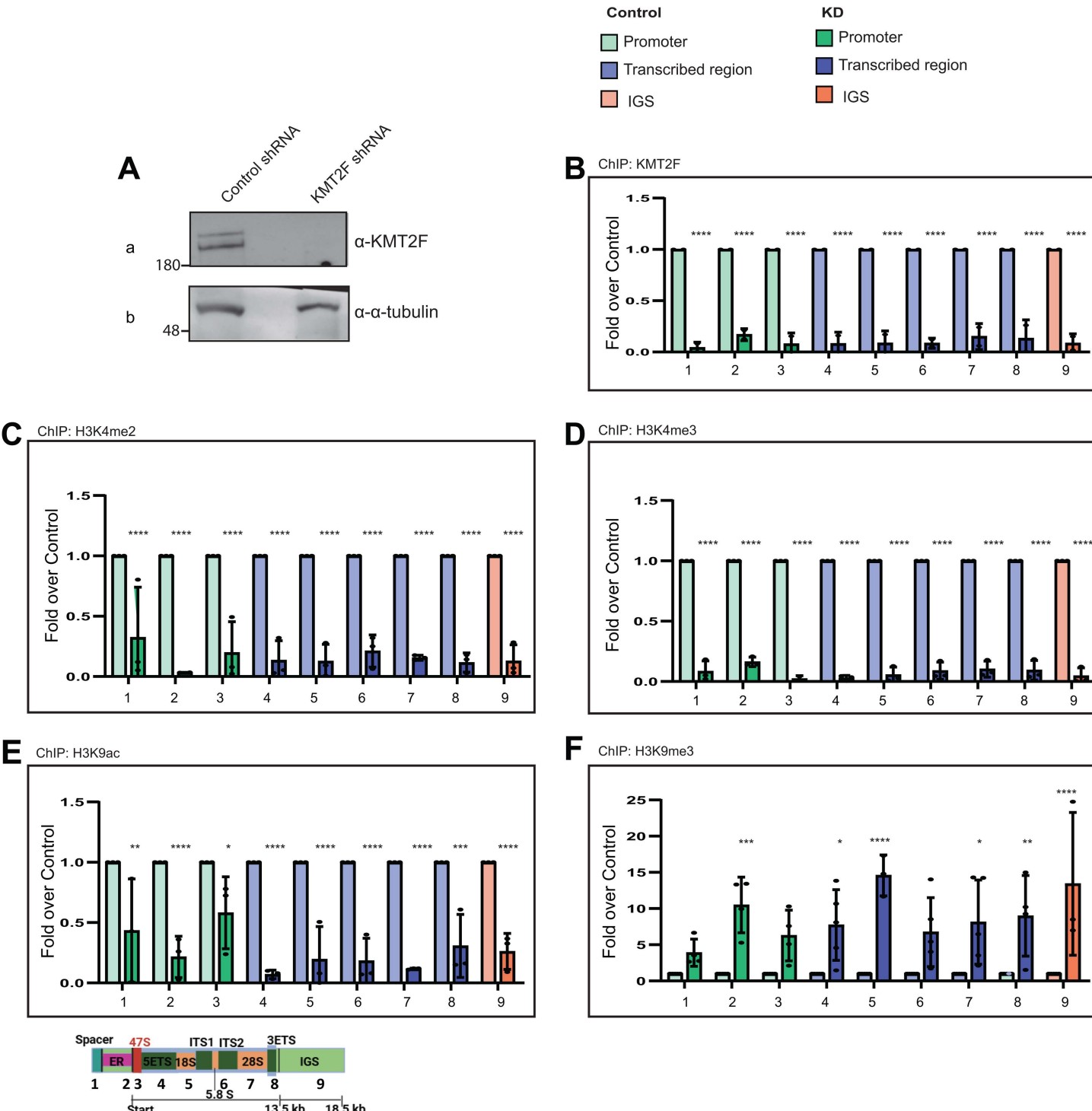

**Fig 5. KMT2F deposits H3K4me3 marks to regulate the epigenetic state of human rDNA. A.** Immunoblots of whole cell extracts were performed on scrambled shRNA (control shRNA) and KMT2F shRNA-treated cells. Immunoblots were probed with α-KMT2F and α-α-tubulin as shown. The uncropped blots are shown in the S1 Raw Images. **B–F.** ChIP qPCR of KMT2F **(B)** H3K4me2 **(C)** H3K4me3 **(D)** H3K9ac **(E)** H3K9me3 **(F)** in control shRNA and KMT2F shRNA conditions. Data is presented as fold over control. Experiments were performed two (for KMT2F) or more than two times as biological replicates. Error bars represent SD. Significance was calculated between control and knockdown conditions for each primer pair using two-way ANOVA with Šidák multiple comparison test. *$P \le 0.05$, ** $P \le 0.005$, ***$P \le 0.0005$, ****$P \le 0.00005$. The underlying values pertaining to B–F can be found in the S1 Data.

first examined KMT2C binding at canonical target loci. KMT2C showed robust enrichment at the known positive control enhancer region of the *PGR* gene, known to be transcriptionally regulated by KMT2C [52], whereas minimal binding was detected at the *Synapsin II* gene, which served as a negative control (S8E Fig). These results confirmed the specificity of the KMT2C ChIP. We next assessed KMT2C occupancy at rDNA using rDNA-specific primers and observed negligible binding of KMT2C under our experimental conditions (S8F Fig), further supporting our findings above that KMT2F does not have a direct role in rDNA transcription. After establishing the absence of KMT2C binding at rDNA, we performed shRNA-mediated depletion of KMT2C and confirmed efficient knockdown by ChIP analysis, which revealed a significant reduction of KMT2C occupancy at the *PGR* enhancer (S8G Fig). No significant changes in either H3K4me2 or H3K4me3 were observed at rDNA following KMT2C depletion (S8H and S8I Fig), consistent with the established role of KMT2C as a mono methyltransferase. Together, these results further validate the specificity of our epigenomic analyses and support the conclusion that KMT2F epigenetically regulates the rDNA transcription.

## Loss of KMT2F HMTs affects the epigenetic landscape of rDNA

H3K4me2 and H3K4me3 marks are associated with positive transcription and accordingly influence the surrounding chromatin epigenetic landscape [53,54]. H3K9 acetylation (H3K9ac) is another mark associated with open chromatin. Therefore, we tested the levels of H3K9ac on rDNA in KMT2F KD. Loss of KMT2F resulted in reduced levels of H3K9ac (Figs 5E and S9A). Consistent with the loss of active marks like H3K9ac and H3K4me3, loss of KMT2F was accompanied by a 5 to 10-fold increase in H3K9me3 mark (Figs 5F and S9B), indicating that by depositing H3K4me3 mark, KMT2F prevents the repression of rDNA and keeps it open for transcription.

## Loss of KMT2F abrogates pre-initiation complex formation by RNA Pol I

Loss of KMT2F decreased rRNA transcription and adversely impacted the rDNA epigenetic landscape. In order to find out exactly what step of RNA Pol I transcription is affected, we undertook further studies. The human rDNA transcription involves the following sequence: rDNA is activated when UBF binds to the RNA Pol I promoter. Then, UBF recruits the SL1 complex to form the PIC at the rDNA promoter. Following this, UBF interacts with RNA Pol I through its RPA34 and RPA49 subunits, while SL1 interacts with the transcription factor RRN3. All factors bind to the 47S promoter as well as the spacer promoter. Once the PIC formation stabilizes RNA Pol I, RRN3 is released from the RNA Pol I complex, and RNA Pol I proceeds with transcript elongation to yield 47S pre-rRNA.

   In KMT2F KD, we observed that the occupancy of RNA Pol I was drastically reduced all across the transcribed region (Fig 6A). Similarly, the binding of both SL-1 complex and RRN3 to the rDNA promoter was highly reduced when compared to control samples (Fig 6B and 6C). We observed a 50% decrease in UBF binding on KMT2F KD on the spacer promoter and 47S promoter, which was further reduced further in the body of the transcribed unit (Fig 6D). Thus, we have shown that KMT2F deposits H3K4me3 marks to keep the rDNA open for transcription (Fig 5). Further, we observe that RNA Pol I pre-initiation complex formation is completely abrogated upon loss of KMT2F, resulting in reduced rRNA transcription. Taken together, our data suggest that H3K4me3 marks deposited by KMT2F are requisite for successful PIC formation and transcription by RNA Pol I at the rDNA locus. As KMT2C did not affect either rDNA transcription or the rDNA epigenetic landscape, we used KMT2C depletion as a negative control for the PIC ChIP analyses. Depletion of KMT2C did not alter either RNA Pol I or UBF occupancy at rDNA in our experimental conditions (S9C and S9D Fig). Together, these findings highlight the distinct roles of individual KMT2 family members in regulating PIC formation at rDNA.

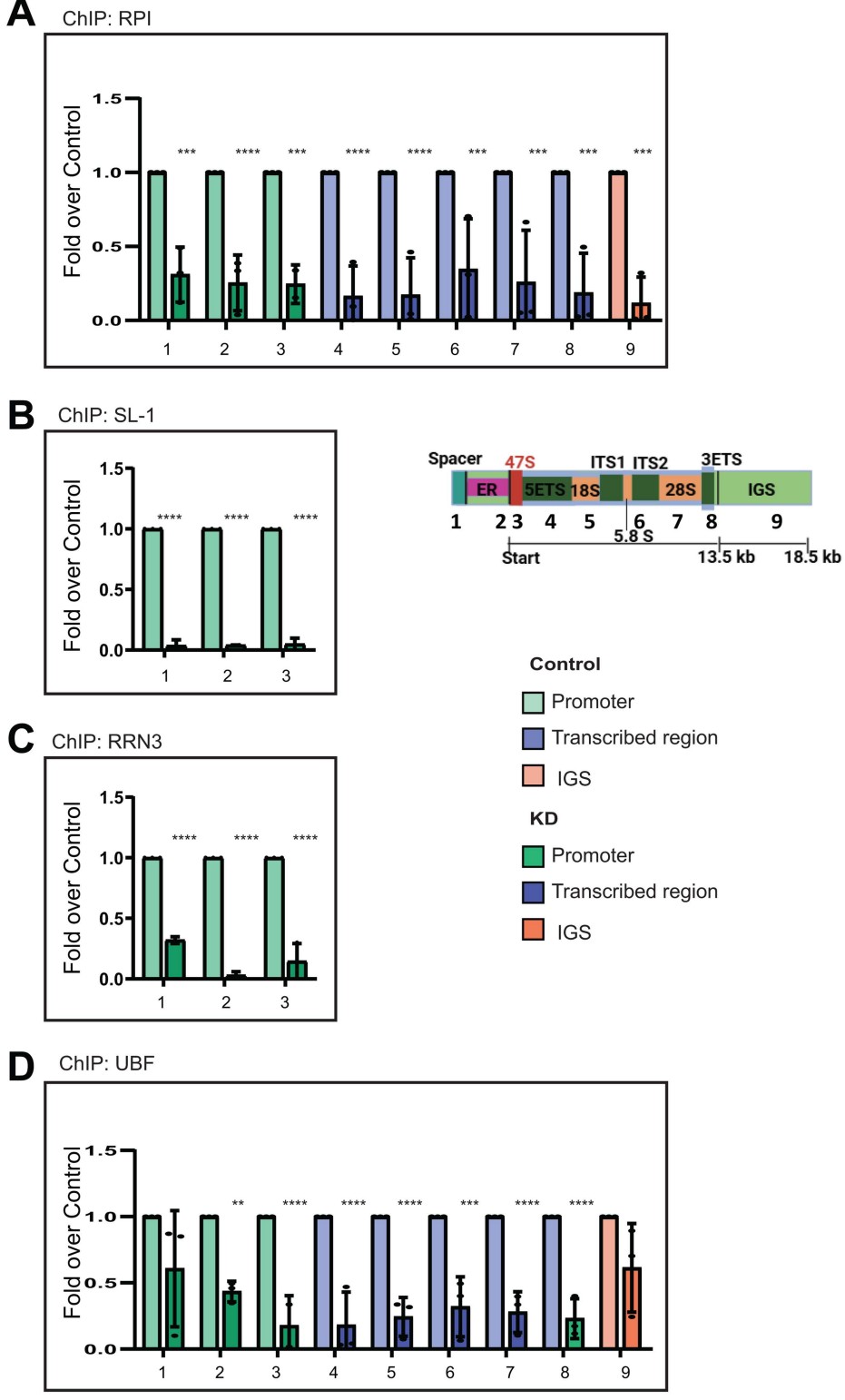

**Fig 6. Loss of KMT2F affects the formation of RNA Pol I pre-initiation complex on rDNA. A–D.** ChIP followed by qRT-PCR of RNA Pol I **(A)**, SL1 represented by TAF1C **(B)**, RRN3 **(C)**, and UBF **(D)** was performed in cells transfected either with scrambled or KMT2F shRNA. Fold over control was

determined by dividing the ChIP signals of KD cells by the ChIP values of control cells for the indicated antibodies. Promoter (#1–3; green), transcribed region (#4–8, blue) and IGS (9; orange) primers were used to quantify ChIP qPCR signals. Error bars represent mean ± SD of two or more individual biological replicates. Significance was calculated between control and knockdown conditions for each primer pair using two-way ANOVA with Šídák multiple comparison test. *$P ≤ 0.05$, ** $P ≤ 0.005$, ***$P ≤ 0.0005$, ****$P ≤ 0.00005$. The underlying values pertaining to A–D can be found in the S1 Data.

### Recruitment of the KMT2F to the rDNA locus

KMT2F lacks the direct DNA-binding domains. However, it is indirectly recruited to the chromatin through its association with various proteins. CFP1 acts as the primary targeting component for the KMT2F complex, recruiting it to CpG island promoters, while WDR82, alone or in association with other factors, targets KMT2F to transcription start sites [40,55,56]. Other recruitment pathways have been described for specific context, like cell cycle-regulation [57]. Hence, KMT2F is engaged in multivalent interactions with the chromatin. As the presence of KMT2F on the RNA Pol I promoter, regulating the PIC formation, bears similarity to KMT2F's binding to transcription initiation sites of RNA Pol II, we decided to explore this further. We hypothesized that the N-terminal RRM (RNA recognition motif) of the KMT2F, which interacts with the WDR82, might be playing a similar role in the recruitment of the KMT2F to the rDNA locus through the WDR82 axis. To test this hypothesis, we cloned the fragment of the RRM spanning 74–172 amino acids in the SFB triple epitope vector, and ectopically expressed it in the cells. In our pull-down experiments with S-protein beads, the RRM showed association with the RNA Pol I, RRN3 and PAF-49 (S9E Fig). Given that RRN3 binds active RNA Pol I specifically at the rDNA promoter, our results support a model in which KMT2F is recruited to the rDNA promoter, most plausibly through an indirect interaction mediated by WDR82.

### The catalytic activity of KMT2F is required for a stable PIC at the rDNA promoter

We have shown the detailed interaction between the KMT2F complex and various components of the RNA Pol I machinery in Fig 3. However, it remains unclear whether the direct physical interaction of KMT2F is responsible for RNA Pol I recruitment to rDNA or if this process depends on the catalytic activity of KMT2F. To investigate this, we conducted affinity pull-down studies in cells ectopically expressing full-length KMT2F or the catalytic dead mutant of KMT2F having a point mutation in the SET domain (KMT2F N1646A). Our S-protein pull-down assays revealed that KMT2F interacts with RNA Pol I subunits as well as RRN3, and this interaction is unaffected by SET domain mutant KMT2F (Fig 7A and 7B). Consistent with our results in S9E, these results show that the interaction of the RNA Pol I complex with the KMT2F is independent of its catalytic activity. As PIC formation is lost upon KMT2F knockdown, we reasoned that maybe its catalytic activity is required to stabilize the PIC on the rDNA promoter. Indeed, when we performed ChIP of RNA Pol I and UBF, on these cells, in the background of KMT2F shRNA, the catalytic dead mutant KMT2F could not sustain PIC formation, unlike the wild type KMT2F (Fig 7C). These results indicated that H3K4me3 deposition by KMT2F is crucial for RNA Pol I PIC formation at the rDNA promoter. Further, they also indicate that KMT2F recruits and stabilizes RNA Pol I PIC formation at rDNA promoter using its catalytic activity to promote transcription of 47S rRNA.

### KMT2A and KMT2F regulate rRNA transcription synergistically

Next, we asked, if KMT2F can promote PIC formation at rDNA promoter, what is the requirement of KMT2A in rRNA transcription? To answer this question, we overexpressed KMT2A alone, KMT2F alone, or both KMT2A and KMT2F. We observed that overexpression of KMT2A alone resulted in a modest 2-fold increase in rRNA transcripts, while overexpression of KMT2F alone showed a 10-fold increase (Fig 7D). Surprisingly, co-expression of both proteins exhibited about a 40-fold increase, indicating that their activities are synergistic to rRNA transcription. Taken together, our results indicate that KMT2F modulates rRNA transcription via its KMT activity while KMT2A acts with KMT2F to augment the activity of RNA Pol I.

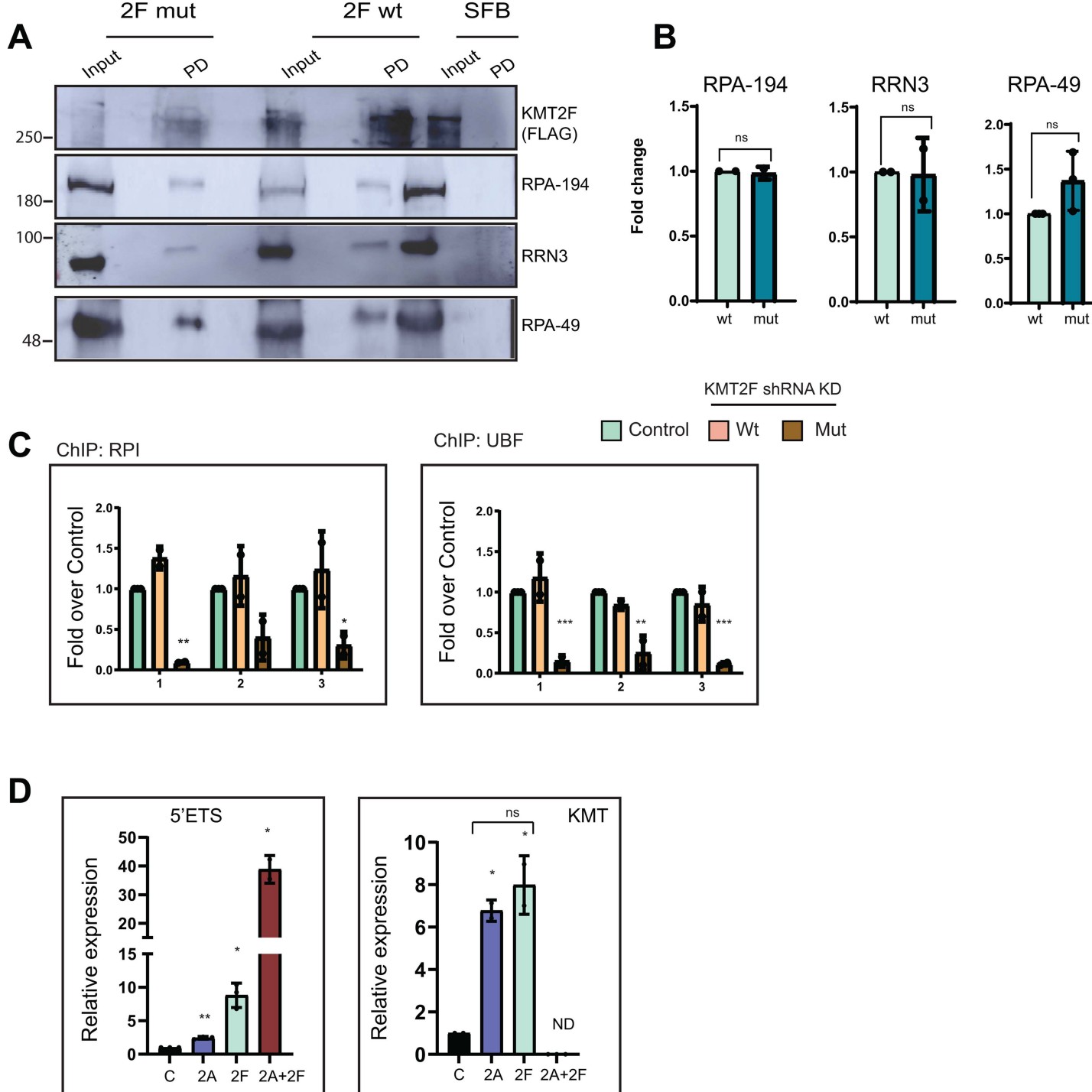

**Fig 7. Catalytic activity of KMT2F is required for the formation of RNA Pol I preinitiation complex on rDNA. A.** The full-length KMT2F (2F wt) and the KMT2F catalytically dead point mutant N1646A (2F mut), along with SFB (mock) were used to study the interaction of the KMT2F with the components of the pre-initiation complex. These constructs were overexpressed in cells and subjected to pulldowns using S-protein beads, detected using anti-FLAG antibody. Immunoblots were probed with various antibodies for the pre-initiation complex (RPA-194, RRN3, and RPA-49) to assess the interaction of KMT2F in both wild-type and mutant conditions. The underlying uncropped blots are shown in the S1 Raw Images. **B.** Represents the quantification of protein interaction levels between KMT2F and the various components of the pre-initiation complex (RPA-194, RRN3 and RPA-49) under wild-type and

mutant conditions, as shown in **A**. The quantification was performed by normalizing the interactions in mutant conditions to those in wild-type conditions. Error bars indicate standard deviation (SD). ns indicates not significant $P > 0.05$, two-tailed Student $t$ test. **C**. Represents the ChIP analysis of the RPA-194 (RNA Pol I) and UBF done in the control cells or in the stable cell lines of the KMT2F expressing KMT2F full length represented as "wt" or in the catalytically dead KMT2F represented as "mut", both the cell lines were treated with 3UTR shRNA targeting the degradation of the endogenous KMT2F. ChIPed DNA in these three conditions was subjected to qRT-PCR and the binding of the RPI and UBF rDNA promoter was detected. Significance was calculated with respect to the control in both the wt and mutant conditions using two-way ANOVA with Šídák multiple comparison test. Error bars indicate SD, *$P \leq 0.05$, **$P \leq 0.005$, ***$P \leq 0.0005$ and ns indicates not significant $P > 0.05$. **D**. KMT2A (2A), KMT2F(2F) were overexpressed alone or together (2A+2F) along with vector control (C) in HEK293 cells. Total RNA was prepared from overexpressed cells followed by cDNA synthesis. 5′ETS transcript levels were quantified in all three cases using primer #3′. The other graph shows the expression levels of KMT2A (2A) and KMT2F (2F) in overexpressed KMT2A and KMT2F HEK293 cells. Data is presented from two or more than two experiments. Significance is calculated between the control and each condition using two-tailed Student $t$ tes*t*. *$P \leq 0.05$, ** $P \leq 0.005$ not significant $P > 0.05$. The underlying values for B–D can be found in the S1 Data.

## Discussion

The H3K4me3 mark, deposited by the KMT2/COMPASS proteins, decorates the transcriptionally active promoters and is widely believed to be associated with activating RNA Pol II-mediated transcription [30]. Here, we show that the H3K4me3 mark deposited by KMT2F shapes the epigenetic landscape of the rDNA and promotes the PIC formation by RNA Pol I for active rRNA transcription. Our results highlight the similarities and differences in how KMT2 enzymes engage with the two RNA polymerases to promote transcription.

### KMT2 complexes interact with RNA Pol I

The three RNA polymerase I-III exist in homeostasis and if the functions of one are affected, they manifest in the transcriptional output of other two polymerases [58,59]. In order to elucidate a direct functional role of KMT2 complexes on the RNA Pol I machinery, and to rule out an indirect effect due to misregulation of RNA Pol II by these complexes, we investigated the protein-protein interaction of COMPASS complex with RNA Pol I components. Our pull-down experiments demonstrated that WDR5 interacted with RNA Pol I subunits as well as with RRN3 and the SL1 complex, factors involved in transcription initiation, as well as with transcription elongation. In support of our findings, WDR5 was identified as a nucleolar protein in human proteome analysis, and depletion of WDR5 has been reported to reduce rDNA transcription [51]. Specifically, we show that WDR5 directly interacts with RPA34 and RPA49, peripheral subunits specific to RNA Pol I. RPA34 and RPA49 form a functional heterodimer at RNA Pol I and have been implicated in RNA Pol I transcription initiation and elongation [46]. Further, they exhibit functional similarity with the TFIIE and TFIIF modules, which regulate RNA Pol II initiation and elongation [60–62]. Consequently, it is plausible that KMTs may modulate RNA Pol I transcription through these interactions. Similarly, we explored the interaction of RNA Pol I-specific subunits with KMT2A and KMT2F. Our endogenous immunoprecipitation experiments revealed that both the N- and C- subunits of KMT2A, as well as KMT2F, effectively precipitated the catalytic RNA Pol I transcriptional component RPA194, as well as UBF, and RRN3. We further mapped the RNA Pol I-interaction domain to RRM of KMT2F. Taken together, our results indicate that the regulation exerted by the KMT2 complexes involves physical proximity with the RNA Pol I complex.

### Unraveling the influence of KMT2 enzymes on the epigenetic landscape of rDNA

The transcription process of rRNA genes necessitates a specialized RNA polymerase (RNA Pol I) and a distinct set of basal transcription factors [63,64]. The basal transcription factors typically include UBF and the SL1 complex (consisting of TBP and TAFs), which play crucial roles in rDNA transcription. Our comprehensive mapping of RNA Pol I and UBF across the human rDNA locus was consistent with their specific binding patterns across the gene body and within the promoter region reported before. Minimal mapping of RNA Pol I was observed in the IGS region, consistent with previous studies [7,37,65]. Earlier research has proposed that epigenetic mechanisms play a role in regulating the transcriptional

activity of rRNA genes. Specifically, histone modifications emerge as pivotal elements in mediating the regulation of rDNA transcription [66]. UBF exhibit a unique pattern of interaction with methylated H3K4, as its binding to rDNA locus is stimulated by H3K4me2 [67]. Moreover, depletion of UBF results in decreased levels of H3K4me3 on the human rDNA promoter [66]. These studies prompted the detailed global bioinformatic analysis of the distribution pattern of histone modifications on rDNA locus. The analysis revealed the distribution of six active histone modifications (H3K4me1, H3K4me2, H3K4me3, H3K9ac, H3K27ac and H3K36me3) and three repressive histone modifications (H3K9me1, H3K27me3 and H4K20me1) across the rDNA locus [7]. Given the pivotal role of KMT2 enzymes in regulating the distribution of H3K4 methylation marks across the genome, we were encouraged to unravel their direct functional significance on the regulation of rDNA locus. We choose KMT2A and KMT2F to conduct ChIP-seq analysis to delineate their distribution patterns on the rDNA locus. Our findings revealed the enrichment of both enzymes on the rDNA locus, with KMT2A exhibiting a preference for the intergenic region, while KMT2F demonstrated a nearly uniform distribution pattern in the RNA Pol I transcribed region. Similar to these KMTs, we found H3K4me2 and H3K4me3 marks enriched on rDNA by ChIP-seq analysis (Figs 2 and S2). To further dissect the specificity of binding of these enzymes and modifications on the rDNA, we performed the ChIP-seq analysis of DOT1L and H3K79me2, which showed negligible binding, thus further building our confidence about the specificity of binding of these enzymes and modifications on rDNA. Collectively, these observations strengthen our confidence in the specificity of the binding patterns reported here and argue against a model in which rDNA simply acts as a generic sink for chromatin modifiers. Rather, our data suggest a regulated and selective epigenetic landscape at rDNA loci, reinforcing the concept that rDNA regulation is governed by distinct chromatin mechanisms within the nucleolar compartment.

To delve deeper into the mechanistic details of how these two proteins regulate the rDNA locus, we tried to generate CRISPR-mediated knockouts of KMT2F. However, as CRISPR-mediated knockout cells of KMT2F were not viable, we were limited to using shRNA-mediated knockdown of KMT2F. Our observations revealed regulation of active signatures (H3K4me2 and H3K4me3) following the loss of this protein. KMT2F depletion exhibited a decrease in H3K4me2, H3K4me3, H3K9ac marks, and a corresponding increase in H3K9me3 mark on promoter and transcribed unit. Therefore, our results reveal that KMT2F-mediated H3K4me3 deposition not only acts directly by regulating RNA Pol I PIC formation but also indirectly, by shaping the epigenetic landscape of the rDNA.

### Drawing comparison between RNA Pol I or RNA Pol II transcription processes and role f KMT2F therein

Although the eukaryotic RNA polymerases differ from each other by performing a vast array of different functions, they share structural similarities in terms of their catalytic core and in the functional steps of the basic transcription process. For instance, despite having 14 subunits (RNA Pol I) and 12 subunits (RNA Pol II), both RNA polymerase still share 7 common subunits. Further, functional similarities, like those exemplified by the heterodimeric complex of RNA Pol I (RPA49-RPA34) functionally resembling the general transcription factor TFIIE-TFIIF modules of RNA Polymerase II, are observed. Similarly, the transcription factor TFIIB plays an analogous role as played by RNA Pol I-associated factor RRN3 in transcription initiation [46]. There are examples of factors that resemble both structurally and functionally in the other processes of transcription, like in the elongation, regulation, etc. [68]. We observe many similarities in how KMT2F engages with RNA Pol I comparable to its engagement with RNA Pol II. Although the KMT2F lacks the direct DNA binding domains, it interacts with WDR82 or CFP1, which targets it to the chromatin through the RNA Pol II or through the CXXC of CFP1. Looking at the broad similarities in the two processes, and our results showing that KMT2F interacts with RNA Pol I components through its RRM domain, we hypothesize a parallel recruitment mechanism for KMT2F to the rDNA locus.

KMT2F plays a critical role in the global deposition of H3K4me3 marks, which have been widely observed to support transcriptional activation of RNA Pol II-regulated genes. H3K4me3 contributes to the formation of the RNA Pol II PIC at active promoters. The TAF3 subunit of TFIID interacts with H3K4me3 to facilitate the recruitment of the TFIID complex, consisting of the TATA-binding protein (TBP) and nearly 14 TBP-associated factors (TAFs) [30]. Additionally, the

H3K4me3 mark can recruit chromatin remodelers, which maintain an open chromatin state at the gene promoters [69]. However, recent reports suggest that H3K4me3 is crucial for RNA Polymerase II pause release [31,32]. Although there is growing evidence supporting the role of H3K4me3 in regulating transcription by RNA Pol II, its role in modulating rDNA transcription by RNA Pol I remains surprisingly unexplored. Our findings indicate that H3K4me3, deposited by KMT2F, is crucial for pre-initiation complex formation on rDNA promoters. To check the specificity of the role played by KMT2F in the formation of PIC on rDNA, and based on our observations of its rDNA-independent role, we used KMT2C as a negative control for our analysis. The absence of rDNA-associated epigenomic signatures, transcriptional effects, and PIC-related activity for KMT2C strongly argues against a non-specific or family-wide role of KMT2 enzymes at rDNA. Instead, these findings highlight a selective and specialized function for KMT2F in orchestrating PIC assembly via H3K4me3 and rDNA regulation within the nucleolar context, thereby substantially strengthening the specificity and biological relevance of our conclusions.

Interestingly, KMT2F physically interacts with RNA Pol I but needs its KMT activity to stabilize the RNA Pol I PIC on the promoter. In contrast to the similarities outlined above between the two polymerase processes, these findings present a divergence, as depletion of KMT2F, or mutations in its SET domain, does not affect RNA Pol II recruitment to the chromatin. However, our results reveal that loss of KMT2F catalytic activity disrupts RNA Pol I recruitment to rDNA. Many factors may contribute to these differences. For instance, recycling and transcription re-initiation by RNA Pol I, dependence of PIC factors like UBF on H3K4 methylation, organization and accessibility of rDNA chromatin, may individually or together necessitate the presence of H3K4me3 on rDNA promoter for PIC formation. This suggests a possible co-dependence of RNA Pol I on KMT2F for proper recruitment at the rDNA, highlighting distinct roles that KMTs play in different transcriptional contexts.

### Impact of KMT2F-mediated rDNA transcription on disease biology

Heterozygous loss-of-function variations in KMT2F have been associated with various pathologies, including childhood speech ataxia [70], early-onset epilepsy [71], and severe developmental disorders [72]. However, the most notable association of KMT2F lies in the development of schizophrenia. Patients diagnosed with schizophrenia have shown increased transcriptional activity of rDNA in both brain tissue and lymphocytes. This heightened activity may arise from elevated rDNA content in the genomes of individuals with schizophrenia compared to healthy controls [73,74]. This proposed interconnection among schizophrenia, KMT2F, and rDNA transcription underscores the importance of further investigating the role of KMT2F in rDNA transcription. Taken together, these studies indicate the necessity for deeper investigation into how KMT2F may deregulate the transcription of the rDNA locus in disease condition, while our study firmly establishes its regulatory role in RNA Pol I transcription.

## Materials and methods

### Cloning

The cDNA sequences of RPA12, RPA34, RRN3, TAF63, and TAF48 were PCR amplified from the cDNA synthesized from total RNA. Subsequently, the amplified cDNA was cloned into the TOPO vector using the TOPO TA Cloning Kit (Invitrogen), and then subcloned into pGEX4T1 to generate GST-tagged constructs. GST RPA43 was generated by cloning the PCR product amplified from TOPO RPA43 (a kind gift from Ingrid Grummt) into pGEX4T1. GST-KMT2A-SET (GST-MLL-D3, [75]) and GST-RPA49 (a kind gift from Ingrid Grummt, [76]) has been described before. RRM was PCR amplified from PCDNA-SFB-KMT2F vector and then sub-cloned in the Xho1 site of the PCDNA-SFB vector. KMT2C targeting shRNAs were designed according to the TRC shRNA design guidelines and cloned into the pLKO.1 vector (Addgene). Briefly, the pLKO.1 vector was digested with AgeI and EcoRI, and the linearized vector was ligated with annealed oligonucleotides encoding KMT2C-specific shRNA sequences.

## Cell culture, stable cell line generation, and transfections

HEK-293 (human embryonic kidney), HeLa (Cervical Cancer cell line), MCF-7 (Breast Cancer cell line), U-2OS (human osteosarcoma), and IMR-90 tert (human lung fibroblast) cells were cultured in DMEM (Gibco), supplemented with 10% (v/v) fetal bovine serum (FBS), 1% (v/v) GlutaMAX, and 100 U/ml penicillin-streptomycin. Authentication of cell lines was conducted via STR profiling (Life Technologies). The cells were maintained at 37 °C in a humidified atmosphere with 5% $CO_2$. Generation of full-length and mutant KMT2A and KMT2F cell lines have been described previously [47,48]. Similar methods were used in the generation of KMT2F full-length and mutant cell lines in the HEK-293 cells [48]. To deplete the endogenous proteins, RNA interference (RNAi) was employed targeting KMT2A, KMT2B, KMT2C, KMT2F, and WDR5 genes, as reported previously [47,48,77]. Briefly, siRNA duplexes were designed and transfected into the cells using Oligofectamine (Thermo Fisher Scientific) as the transfecting agent. A siRNA sequence targeting the firefly luciferase gene was used as a control. KMT2F and KMT2C shRNAs were transiently transfected into HEK-293 cells using PEI, as described previously [48]. Scrambled shRNA was used as a control. Cells underwent two rounds of transfections at a time interval of 24 hours and were collected after 72 hours. Two shRNAs targeting the gene (#1) and 3′UTR (#2) were used to knock down KMT2F to perform ChIP experiments [48], while only shRNA #2 was used in experiments of Fig 7C to protect complementing KMT2F cDNAs.

## Protein expression and purification

All GST-tagged proteins were expressed using either BL 21 (DE3) or Rosetta Gami DE3 *Escherichia coli* strain (Novagen). Protein expression was induced by adding 0.1 mM Isopropyl β-D-1-thiogalactopyranoside (IPTG) for 6 hours at 18°C when the culture reached 0.6 OD. The cells were then pelleted and lysed in lysis buffer (50 mM Tris pH 7.5, 150 mM NaCl, 0.5% NP-40, and 1 mM PMSF), followed by incubation with glutathione agarose beads (Sigma) at 4°C for 2–3 hours. Subsequently, the beads underwent washing with ice-cold lysis buffer 3–4 times. Protein concentration was determined via SDS-PAGE followed by CBB staining.

## Co-immunoprecipitation, GST affinity pull-downs, and immunoblot analysis

Mammalian cell lysates were prepared by either performing whole cell lysis using NETN (100 mM NaCl, 20 mM Tris-HCl pH 8, 0.5 mM EDTA, and 0.5% Nonidet P-40) or specifically conducting nuclear lysis [78]. Cell/nuclear lysates were treated with a cocktail of protease inhibitors (Sigma) and was used for immunoprecipitation/pull-down experiments. The pre-cleared lysate was incubated with 1μg of IgG, KMT2A, or KMT2F antibodies, respectively, or with S protein beads for SFB-tagged proteins at 40 °C O/N. Following incubation with antibody, the cell lysate for endogenous pulldowns was incubated with 20 ml of 50% slurry of Protein A beads for 1 hour at 40 °C. The pelleted Protein A/S protein beads were washed 3 times with IP wash buffer (15 mM TrisCl pH 7.5, 100 mM KCL, 0.02%NP-40) and boiled in Laemmli buffer. The samples were run on a SDS–PAGE followed by immunoblotting. For pull-down experiments involving GST-tagged proteins, an equal amount of purified bead-bound proteins were incubated with HeLa or HeLa spinner whole-cell or nuclear extracts for 2–3 hours, followed by washing with IP wash buffer. The boiled protein extracts were separated by SDS–PAGE and transferred to either nitrocellulose (Amersham, 10600003) or PVDF (Amersham, 10600023) membranes. Endogenous KMT2A-C subunit (A300-374A, Bethyl Labs), KMT2A-N subunit (A300-086A, Bethyl Labs), and KMT2F (A300-289A, Bethyl Labs) were pulled down and immunoblotted with RPA194 (SC-48385, Santa Cruz), UBF (SC-13125, Santa Cruz), RRN3 (Ab-112052, Abcam), RNA Pol II (ab252855, Abcam), and KMT2A or KMT2F antibodies mentioned above. Similarly, SFB pull-downs were probed with RPA-49 (HPA022-527, Sigma), RPA194 (SC-48385, Santa Cruz), RRN3 (Ab-112052, Abcam) or with FLAG (F7425, Sigma). Blots were scanned using an Odyssey infrared imager (LI-COR) or ImageQuant LAS 500.

## In-vitro interaction studies

Thrombin enzyme (Sigma) was utilized to cleave the GST tag in purified GST-WDR5 at 22 °C for 18 hours. The intact WDR5 protein post-cleavage was retrieved and employed for in-vitro interaction studies. Subsequently, bacterially purified GST tag proteins were subjected to incubation with purified WDR5 (500 ng each) in interaction buffer (IP dilution buffer: 50 mM TRIS pH 7.4, 150 mM NaCl, 0.1 mM EDTA, 0.1% NP40, 100 μg/ml BSA also containing 200 mM KCl) for a duration of 12 hours. Beads were collected by centrifugation, followed by thorough washing with IP wash buffer. Finally, Western blotting was employed to identify bound protein complexes.

## Immunofluorescence imaging

Cells (U-2OS, MCF-7, HeLa, HEK293, and IMR90-tert) were cultured on polylysine-coated glass cover slips. Different fixation protocols were employed to detect the nucleolar localization of endogenous KMT2A and KMT2F proteins. For KMT2A, cells were fixed with pre-chilled acetone for 90 s at −20 °C for HEK293, and 4 min in all other cell lines. KMT2F fixation was achieved with pre-chilled acetone for 90 s at −20 °C for HEK293. While cells were incubated with pre-chilled methanol and acetic acid (1:1) for 4 min for all other cell lines. To verify the exogenous expression of KMT2A, KMT2F and their mutants, we fixed cells using 4% paraformaldehyde for 10 min, followed by permeabilisation using 0.2% Triton X-100 (Amresco, 0694) as described before (S7A Fig; [77]). Subsequently, fixed cells were blocked with one percent BSA for 30 min at room temperature. The cells were then incubated overnight at 4 °C with antibodies against KMT2A-C/N (1:100) (A300-374A/A300-086A Bethyl Labs), KMT2F (1:100) (A300-289A, Bethyl Labs), FLAG (F7425, Sigma) while B23/nucleophosmin (1:150) (B0556, Sigma), Fibrillarin (1:400) (ab 5821, Abcam) or UBF (1.400) (SC-13125, Santa Cruz) were used as a marker for the nucleolus. Following primary antibody incubation, the cells were washed and incubated with Alexa 488 (1:1,000, Thermo Fisher Scientific A11034), Alexa 633 (1:500, Thermo Fisher Scientific A21050), and/or Alexa 594 (1:500, Thermo Fisher Scientific A11030) conjugated anti-rabbit or anti-mouse secondary antibodies at room temperature for 1 hour. The samples were mounted in VECTASHIELD Mounting Medium (Vector Laboratories-H1200) with 4-6-diamidino-2-phenylindole (DAPI) to stain the DNA. Images were acquired using a ZEISS LSM 900 inverted confocal microscope with a 63×/1.4 oil immersion lens.

## RNA isolation and qRT-PCR

Total RNA isolation and cDNA preparation was done as described [48]. Before cDNA synthesis, the RNA was treated with TURBO DNase (Thermo Fisher Scientific) for 30 min at 37 °C to eliminate genomic DNA contamination. To ensure RNA purity, a no-enzyme RNA amplification step was conducted prior to cDNA synthesis. After confirming the absence of DNA contamination, cDNA synthesis was carried out using the SuperScript III Reverse Transcriptase kit (Thermo Fisher Scientific) following the manufacturer's instructions. RT-qPCR analysis was conducted using either the 7500 Real-Time PCR (Applied Biosystems), QuantStudio 5 Real-Time PCR System (Applied Biosystems), or Bio-Rad (CFX-maestro) platforms, employing the DyNAmo ColorFlash SYBR Green qPCR kit (Thermo Fisher Scientific). Transcript levels were quantified using the $2^{-\Delta\Delta Ct}$ method [79]. Primer sequences are provided in S1 Table.

## [32P] metabolic labeling and nascent ribosomal RNA quantification

Cells were incubated in phosphate free media containing 10% FBS for 2 h followed by incubation with DMEM containing 0.1 mCi/ ml of [32P] orthophosphate for 1 h. Labeled RNA was extracted by TRIzol (Ambion) reagent. RNA was quantified with NanoDrop, divided into two portions for each sample from different treatments, and one portion was loaded on 1.2% agarose gel and ran at 60 V/cm for 5 h. Gel was further dried at 8 °C for 1 h. RNA was visualized by autoradiography (Typhoon FLA-9500, GE) and densitometry quantification was done by Phosphorimaging. 45S rRNA levels were quantified between the control and treatment by normalizing 45S RNA levels with 28S rRNA levels. To visualize 28S rRNA, second portion of the sample was run on denaturing agarose gel containing nine percent formaldehyde and visualized after staining with ethidium bromide.

## Chromatin immunoprecipitation

ChIP experiments were performed as previously described by [48,80]. using the following antibodies: KMT2A-C (A300-374A, Bethyl Labs or in-house as described in [77] KMT2F (A300-289A, Bethyl Labs), H3K4me2 (ab32356, Abcam); H3K4me3 (ab8580, Abcam or 07-473 Millipore); H3 (ab1791, Abcam); H3K9me3 (ab8898, Abcam); H3K9Ac (ab4441, Abcam); RPA194 (SC-48385, Santa cruz); UBF (SC-13125, Santa cruz); RRN3 (Ab-112052, Abcam); KMT2C (SC-130173, Santa cruz) and TAF1C (A303-698A, Bethyl labs);. The relative occupancy or percent input of the immunoprecipitated protein at ribosomal DNA locus was estimated by RT-qPCR as follows: $100 \times 2(Ct\ Input - Ct\ IP)$, where Ct Input and Ct IP are mean threshold cycles of RT-qPCR on DNA samples from input and specific immunoprecipitations, respectively. To measure fold over control, fold change over the ChIP values obtained in the control cells was used. Primer sequences are provided in S1 Table.

## ChIP-seq and analysis

The ChIP DNA (10 ng) of KMT2A and KMT2F from HEK-293 cells was utilized to construct ChIP-sequencing libraries employing the NEB Next Ultra II DNA Library preparation kit. Subsequently, paired-end sequencing (2 × 150 bp) was carried out on the Illumina Nextseq 2000 platform, utilizing the sequencing services provided by the CDFD sequencing facility (National Genomics Core (NGC), CDFD. The ChIP-seq datasets of KMT2A and KMT2F, as well as previously published ChIP-seq datasets of H3K4me2 (GSM5954234), H3K4me3 (GSM5954235), Input for H3K4me2 and H3K4me3 (GSM5954242) [43] and UBF (SRX035783), RNA Pol I (SRX035784). Input for UBF and Pol I (SRX035785) [7] for WDR82 (GSE186758) [41] and for WDR5 (GSE60897) [39], for H3K79me2 (GSM4503592), [81], and DOT1L (DRA004872) [42] were analyzed to assess their binding on human rDNA, using a pipeline previously described for rDNA [33]. Briefly, the quality of raw sequence data from experimental (IP) and input DNA samples was assessed using FastQC version 0.11.4 (https://www.bioinformatics.babraham.ac.uk/projects/fastqc/). Adapter removal from raw sequencing read pairs was performed using TrimGalore (v0.6.7) (https://github.com/FelixKrueger/TrimGalore) or Trimmomatic. The trimmed sequencing reads were then mapped to the customized human reference genome (hg38-rDNA) using indexed Bowtie2 (version 2.3.2). SAM files obtained after alignment were converted to BAM format using Samtools (version 1.9). Sorting and indexing of BAM files was done using Samtools (version 1.9). Visualization tracks were created using DeepTools (BamCoverage). Subsequently, BigWig (BW) files generated for input and experimental data (IP) post- BamCoverage were compared using BW Compare (https://deeptools.readthedocs.io/en/develop/content/tools/bigwigCompare.html). The individual BW files for each dataset, generated after comparison, were subsequently visualized using IGV (Integrated Genomic Viewer).

## Differential distribution of ChIP-seq signals at rDNA and genome-wide regions

To compare ChIP-seq signal distribution between rDNA and the rest of the genome, the annotated human rDNA assembly (hg38-rDNA) was segmented into promoter, gene body, and intergenic regions using hg38-rDNA-specific annotation files. Strand-specific BED files were generated for each feature, including promoter (+/−), gene body (+/−), and intergenic (+/−) regions. Aligned and normalized BAM files were quantified over these BED regions using the multiBamCoverage function from deepTools (v3.x) with default parameters, and normalized read counts were obtained for each genomic feature. Read counts for each segmented region was normalized to their respective genome size and were then expressed per kilobase of genomic length. ChIP-seq reads mapping to rDNA originate from approximately 400 tandem rDNA repeats in the human genome, normalized read counts were further divided by 400 to estimate the average signal per individual rDNA repeat. Feature-specific enrichment was subsequently calculated as the percentage of total mapped reads corresponding to promoter, gene body, intergenic, and rDNA regions. The relative distribution of reads across these genomic features was visualized using pie charts, enabling direct comparison of chromatin occupancy across rDNA regulatory and structural regions.

## Generating read counts for each primer across the rDNA locus

To compare ChIP-seq signals at the rDNA locus with ChIP-qPCR results, we performed a quantitative analysis of read counts corresponding to each qPCR primer region. BED files were generated by defining the genomic coordinates of each primer amplicon based on the hg38-rDNA assembly, with strand orientation assigned as sense (+) or antisense (−) according to primer alignment. These BED files were used to quantify ChIP-seq signal from normalized BAM files using the multiBamCoverage function from deepTools (v3.x) with default parameters. The resulting read counts for each primer region were converted to RPKM (reads per kilobase per million mapped reads) using the following formula:

$$RPKM = (\text{Reads mapped to each primer region} \times 10^3 \times 10^6)/$$
$$(\text{Amplicon length} \times \text{Total number of mapped reads genome wide}).$$

## RNA-seq analysis

RNA-seq analysis of KMT2A and KMT2F was performed under control and knockdown conditions as described previously [48], with minor modifications. Briefly, read counts were quantified using feature Counts [82]. Differential gene expression analysis was conducted using DESeq2, an R Bioconductor package. *P*-values were adjusted using the Benjamini–Hochberg method, and genes with an adjusted *P*-value <0.05 and a fold change ≥1.0 were considered differentially expressed. Ribosome-related Gene Ontology (GO) associated with ribosomal structure and biogenesis terms were obtained from the Gene Ontology Annotation (GOA) database, and gene set enrichment analysis was performed using GSEA (R Bioconductor package). All analyses were performed using in-house scripts written in R, Perl, and Python.

## Statistical analysis

Statistical analyses were performed using GraphPad Prism 9.3 software. Student *t* test and one or two-way ANOVA were applied as specified in the figure legends. Statistical significance was determined by comparing the mean of the control group with that of the test group. Error bars indicate the standard deviation (SD).

## Supporting information

**S1 Fig. Endogenous KMT2A and KMT2F localize to the nucleolus: Shows immunofluorescence (IF) of KMT2A and KMT2F under control and depletion conditions in U2OS and HEK293 cells, as well as in additional cell lines.**
(PDF)

**S2 Fig. ChIP-seq analysis of human rDNA: Depicts the ChIP-seq analysis of KMT complex proteins along with the associated epigenetic modifications on the whole human rDNA locus.**
(PDF)

**S3 Fig. KMT2 and RNA Pol I bind to the human rDNA repeats: Confirms the binding of selected PIC proteins in HEK293 cells and demonstrates the occupancy of KMT2A/KMT2F along with H3 on rDNA.**
(PDF)

**S4 Fig. Correlation between ChIP-seq and ChIP-qPCR data: Shows read counts for primer regions derived from ChIP-seq data, together with their enrichment scores obtained by qRT-PCR on human rDNA.**
(PDF)

**S5 Fig. Profiling of KMT2A and KMT2F-regulated genes reveals a coordinated epigenetic and transcriptional regulation of rDNA transcription: Shows quantitative analysis of ChIP-seq data for KMTs, histone modifications, and**

**PIC occupancy on rDNA.** The second part describes the effect of KMT2A and KMT2F depletion on ribosome biogenesis as assessed by RNA-seq.
(PDF)

**S6 Fig. RNAi-mediated depletion of KMTs affects ribosomal DNA transcription: Shows the transcriptional analysis in the depleted conditions of KMT2A/F on the rDNA.**
(PDF)

**S7 Fig. Depletion of KMTs affects ribosomal DNA transcription in different cell lines: Shows the transcriptional analysis in the depleted conditions of KMT2s on the rDNA.**
(PDF)

**S8 Fig. KMT2F deposits H3K4me3 marks to regulate epigenetic state of human rDNA: Shows the effect on the rDNA epigenome in the KMT2C-depleted conditions and in the KMT2F-depleted conditions on the canonical target genes.**
(PDF)

**S9 Fig. PIC formation under KMT2C depletion conditions: Shows the effect on the PIC formation in the KMT2C-depleted conditions.**
(PDF)

**S1 Table. Lists the sequences and genomic coordinates of human rDNA primers, along with ChIP primers for canonical target genes, primers used for transcriptional assays, and shRNA sequences.** References for each primer set are also provided.
(PDF)

**S1 Data. Underlying data for** Figs 1–7**.** Excel file containing individual data points used for graphs made in this manuscript.
(XLSX)

**S1 Raw Images. Original uncropped blots used in this study.** A PDF file containing all uncropped Western blots or gel images used in this manuscript.
(PDF)

## Acknowledgments

We thank Ingrid Grummit for providing the RPA43 and RPA49 constructs. We are thankful Z.U. Zargar for assistance with initial ChIP experiments, K. Malik for assistance with siRNA blots, and A. Pandiyan, N. Velugonda, A. Mahato, A. Gupta, and D. Saikia for their assistance with RNA/ChIP-seq analysis.

## Author contributions

**Conceptualization:** Amit Mahendra Karole, Shweta Tyagi.

**Formal analysis:** Kaisar Ahmad Lone.

**Funding acquisition:** Shweta Tyagi.

**Investigation:** Kaisar Ahmad Lone, Amit Mahendra Karole, Geetanjali Ravindran.

**Methodology:** Kaisar Ahmad Lone, Amit Mahendra Karole, Geetanjali Ravindran.

**Resources:** Amit Mahendra Karole.

**Validation:** Amit Mahendra Karole.

**Writing – original draft:** Kaisar Ahmad Lone.

**Writing – review & editing:** Shweta Tyagi.

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
