## [Editor Report · Decision Letter 0]

17 Mar 2025

Dear Dr Tyagi,

Thank you for submitting your manuscript entitled "H3K4me3 methyltransferase KMT2F promotes pre-initiation complex formation by RNA Polymerase I to regulate ribosomal RNA transcription." for consideration as a Research Article by PLOS Biology. Please accept my sincere apologies for the delay in getting back to you as we consulted with an academic editor about your submission.

Your manuscript has now been evaluated by the PLOS Biology editorial staff, as well as by an academic editor with relevant expertise, and I am writing to let you know that we would like to send your submission out for external peer review.

Once your full submission is complete, your paper will undergo a series of checks in preparation for peer review. After your manuscript has passed the checks it will be sent out for review. To provide the metadata for your submission, please Login to Editorial Manager (https://www.editorialmanager.com/pbiology) within two working days, i.e. by Mar 19 2025 11:59PM.

Kind regards,

Richard

Richard Hodge, PhD

rhodge@plos.org

PLOS

---

## [Decision Letter · Decision Letter 1]

21 May 2025

Dear Shweta,

Thank you for your continued patience while your manuscript "H3K4me3 methyltransferase KMT2F promotes pre-initiation complex formation by RNA Polymerase I to regulate ribosomal RNA transcription" was peer-reviewed at PLOS Biology as a Research Article. Please accept my sincere apologies for the delays that you have experienced during the peer review process. Your manuscript has been evaluated by the PLOS Biology editors, an Academic Editor with relevant expertise, and by two independent reviewers.

As you will see in the reviewer reports, which can be found at the end of this email, although the reviewers find the work potentially interesting, they have also raised a substantial number of important concerns. Based on their specific comments and following discussion with the Academic Editor, it is clear that a substantial amount of work would be required to meet the criteria for publication in PLOS Biology. However, given our and the reviewer interest in your study, we would be open to inviting a comprehensive revision of the study that thoroughly addresses all the reviewers' comments. Given the extent of revision that would be needed, we cannot make a decision about publication until we have seen the revised manuscript and your response to the reviewers' comments. Your revised manuscript would need to be seen by the reviewers again, but please note that we would not engage them unless their main concerns have been addressed.

As you will see, Reviewer #1 is generally positive and thinks the findings are interesting. However, the reviewer notes that additional experiments should be included to to strengthen the claim that KMT2A/F, PIC and H3K4me2/3 are enriched at the rDNA locus, as well as more directly showing that KMT2F loss modulates the epigenomic landscape. Reviewer #2 is more critical and raises concerns that the manuscript lacks important controls and that several experimental revisions are required to fully support the conclusions. This includes ruling out that broad transcriptional re-programming explains the decrease in rRNA transcription in cells lacking KMT2A/F, controlling for whether KMT2A/F loss affects cell viability/growth, including positive controls for the co-IPs and repeating experiments consistently in the same cell lines.

We appreciate that these requests represent a great deal of extra work, and we are willing to relax our standard revision time to allow you 6 months to revise your study. Please email us (plosbiology@plos.org) if you have any questions or concerns, or envision needing a (short) extension.

**IMPORTANT - SUBMITTING YOUR REVISION**

*Resubmission Checklist*

*Published Peer Review*

*PLOS Data Policy*

*Blot and Gel Data Policy*

Best regards,

Richard

Richard Hodge, PhD

rhodge@plos.org

REVIEWS:

Reviewer #1: The manuscript entitled "H3K4me3 methyltransferase KMT2F promotes pre-initiation complex formation by RNA Polymerase I to regulate ribosomal RNA transcription" by Lone et al. reveals a role for this enzyme in controling the epigenomic landscape and transcriptional activity at the rDNA locus.

While much is known about how histone modifications control RNA Pol II activity throughout the genome, relatively less is know on RNA Pol I regulation. The findings are thus interesting and novel, and should be of interest to a broad community. While the he data presented is overall convincing, I would advice the authors to consider the points below to strengthen their message before publication.

Specific comments:

1) Figure 1A: In contrast to what the authors state (page 16), KMT2A appears nucleolar but KMT2F appears peri-nucleolar. The distribution of DAPI signal is also very different in the two snapshots. Can this be in any way related to the somewhat distinct genomic distribution profiles of these proteins shown in Figure 1B-E?

2) Figures 1 and 2: while the custom assembly genome used in this study has been validated by previous reports, as rDNA is a repeat genomic segment, it is difficult to quantitatively interpret the observed enrichments compared to the rest of the genome. The authors could strengthen their observation that KMT2A/F, PIC and H3K4me2/3 are enriched at the rDNA locus by:

- Analyzing their ChIP-seq data on the now available Telomere-to-Telomere (T2T) genome build.

- In light of these analyses, more thoroughly illustrate where else in the genome KMT2A and more importantly KMT2F bind, and quantify how specific binding of the KMT2s is to the rDNA locus relative to the rest of the genome (a semi-quantitative attempt to do this is presented in the qPCR experiments, but this could more generally quantified genome-wide by exploiting the seq data).

- Directly comparing the distributions of KMT2A/F with H3K4me2/3, which would show, it appears, that the histone marks correlate with the distribution of KMT2F but not KMT2A at the rDNA locus.

- Providing negative controls. For Figure 1, this could be the distribution analysis of one KMT2 family protein and/or associated factor that is NOT enriched at rDNA. For Figure 2 this could be the distribution of a histone modification that is not enriched at rDNA (H3K4me1 for example? or other lysine methylations such as K27me?)

3) In Figure 3, the the sub-panels "a" and "b" shown in several main panels are not clearly explained. Could the authors comment on these?

4) In Figure 3A, the information provided in the cartoon regarding the interaction between KMT2A/F and the PIC is inferred by the authors from the results in Figures 3-5. For clarity, I would advice the authors to show their model figure after the results that support it and not before.

5) Figure 4: In the main text, references to SuppFig4G and 4F are inverted.

6) In Figure 4, the authors measure 45S rRNA following RNAi treatment against different KMT2 proteins, and conclude that KMT2B but not KMT2C control rRNA levels. However, the authors do not show evidence that their RNAi experiments are working (i.e that the KMT2B and C indeed display reduced levels as expected). Also, why did the authors exclude KMT2D from their analysis? And how specific do they think is the effect of KMT2A/F on rDNA if 3 out of 4 tested KMT2s have similar effects? In light of this data, it is difficult to understand why the study (as highlighted by the manuscript title) focuses on KMT2F only.

7) In Figure 5, one assumes that the shRNA tool targeting KMT2F used in this figure was not used in previous figures because it was incompatible with rescue constructs? The authors should clarify this point.

8) Figure 5 and 6 show a striking effect of KMT2F loss on the epigenomic landscape at the rDNA locus and the formation of the pre-initiation complex. This is the most important finding in this study, and deserves to be strenghtened. Furthermore, an additive/synergistic effect of KMT2A and F overexpression on rRNA expression is shown in Figure 6, it would have been important to evaluate the status of the epigenome and PIC. To do so the authors could:

- Evaluate the epigenomic and PIC status on KMT2A knockdowns as well as control knockdown (KMT2C for example).

- Evaluate the epigenomic and PIC status upon (co-)overexpression of KMT2A/F.

- Evaluate the impact on the epigenomic and PIC status upon rescue with mutated forms of KMT2F (if possible, considering the construct sequences).

9) Two paragraphs of the main text refer to a supplementary figure (7). These results are convincing and interesting, although negative. For clarity, I would advice the authors either include this information as a main Figure or reduce the respective text.

Reviewer #2: Comments for PBIOLOGY-D-25-00691_R1

In this manuscript titled “H3K4me3 methyltransferase KMT2F promotes pre-initiation complex formation by RNA Polymerase I to regulate ribosomal RNA transcription,” the authors aimed to identify a role for H3K4me3 and its methyltransferases in transcription of ribosomal DNA (rDNA). Previous groups demonstrated that H3K4me2/3 is present at rDNA so the authors sought to test 1) which H3K4 methyltransferases deposit these modifications, and 2) whether loss of these methyltransferases disrupt rDNA transcription. Using immunofluorescence, they identified KMT2A and KMT2F localize to the nucleolus and, using chromatin immunoprecipitation, bind rDNA. Knockdown of KMT2A or KMT2F decreased transcription of rRNA and occupancy of RNA polymerase I (RNAP1) at rDNA. Using protein pulldowns or immunoprecipitations, they show that KMT2A and KMT2F interact with regulators of RNAP1 and they propose a domain of KMT2F that facilitates the RNAP1 interaction. Lastly, they show that KMT2F’s methyltransferase activity is not required for interacting with RNAP1 but is necessary for RNAP1 transcription.

The authors have shown that KMT2A and KMT2F play a role in the transcription of mammalian ribosomal RNA (rRNA). The authors show colocalization with B23, and these proteins interact with core RNA Pol I machinery and are likely present at the rDNA. The authors also show that loss of KMT2F decreases rRNA transcription. Impressively, the authors also show that overexpression of KMT2A and KMT2F increases 5’ETS levels.

However, the manuscript is lacking several critical experiments (including controls) that hamper the interpretation of the data. There are several experiments that provide conflicting or inconsistent data, which was ignored. While there are some compelling data provided, the manuscript needs extensive revisions, including many additional experiments, to confirm their claims or reconsider their results.

Major points

1) In the introduction, the authors state that KMT2A and KMT2F deposit H3K4me2/3 at gene promoters, and that loss of these enzymes and modifications “leads to a widespread decrease in transcriptional output” by RNA polymerase II (RNAP2). Then, the simplest explanation of their data is that loss of KMT2A/F leads to a decrease in the expression of factors that regulate rRNA transcription and/or processing, meaning the decrease in rRNA transcription is not by direct epigenetic regulation at rDNA. Can the authors show that there isn’t broad genic transcriptional re-programming that can explain the decrease in rRNA transcription in cells lacking KMT2A/F?

2) rRNA transcription is strongly linked to cell proliferation. The authors need to show whether KMT2A/F loss affects cell viability/growth.

3) For Figures 3, 4 and 5 and several supplemental figures, Western blots validating shRNA efficacy and transgene re-expression are essential to interpreting this data. Also, it is important to also show H3K4me2/3 levels in these different cell lines.

4) For co-immunoprecipitations and pulldowns, it would strengthen the authors arguments/model to show positive controls such as RNAP2 to show how strong the interactions are relative to known binders.

5) While the use of different cell lines is typically helpful to interpret the significance of the data, here many datasets are compared across cell lines, but all cell lines have different epigenetic backgrounds, proliferation rates, and immortalization methods, each of which can affect rDNA transcription. For example, their blots in Figure 3 were done largely using HeLa lysates to demonstrate KMT interactions with RNA Pol I machinery, but genomic and PCR studies were done largely in HEK293 or U2OS cells. Additionally, in Figure 1 and Figure S1, KMT2A and KMT2F localizations appeared cell line-dependent, but the only validation for their localization was done in U2OS cells (Fig. S1b). To increase confidence in these phenotypes, it will be important to repeat some of these experiments consistently in the same cell lines.

6) For many figures, inappropriate statistical tests were used.

Minor points

1) The nucleolar localization of KMT2s in supplemental figure 1 is not convincing. This is also concerning because this looks very different than what is shown in main Figure 1.

2) The authors state “We found that WDR82 showed higher enrichment in the transcribed unit (Supplementary Figure S2), consistent with KMT2F binding.” But that is a vast overstatement based on the data. The presence of WDR82 at rDNA does not mirror KMT2F strongly at all. And you certainly can’t state that KMT2F binds WDR82 from this data.

3) The ChIP-PCR and ChIP-seq results don’t entirely align, and it is not clear at all how or what statistical tests are use for all ChIP-PCR experiments.

4) For H3K4me2/3 in Fig. 2, it is hard to tell whether there is an enrichment at rDNA or whether the mark is simply present at rDNA. Meaning, is this mark enriched relative to other genomic loci? Also, while these marks look similar from ChIP-seq but not by ChIP-PCR. How can you reconcile this?

5) The authors state “Again, we validated our findings by ChIP analyses and observed consistently high enrichment of these marks on RNA Pol I promoter and IGS region in HEK293(Figure 2B-C).” However, this should read “rDNA promoter”

6) For Fig. 3C, It is hard to interpret the data because the KMT2A-SET positive control and input are on a separate blot, making it difficult to interpret how strongly WDR5 binds to RPAs.

7) For Fig. 3E, is the interaction with RPOL1 components dependent on solely KMT2A or KMT2F? Can you do co-IPs in KMT2A or KMT2F siRNA?

8) For Fig. 4A, the authors state “The transcript levels of KMT2A, KMT2F and WDR5 decreased by more than 60%...” but they need to show protein levels.

9) The authors state “Remarkably, consistent with the role of SET domain of KMT2F in our rRNA transcript analyses, knock down of KMT2F resulted in drastic reduction of H3K4me2 and H3K4me3 levels on the whole rDNA loci (Figure 5C-D, S6B-C),” but how are the global levels of H3K4me2/3 affected? Have you performed western blots or ChIP-seq or PCR at other loci?

10) The results from Fig. 5C are difficult to interpret because the ChIP seq and ChIP PCR are not concordant. Also, the three biological replicates shown are very inconsistent. To definitively show this, you’d need to also look at K4me2/3 levels in KMT2A depleted cells.

11) The authors state “….indicating that by depositing H3K4me3 mark, KMT2F prevents the heterochromatinization of rDNA and keeps it open for transcription.” This is an over interpretation of the data given what is shown.

12) Based on the results of Fig. S7C, the authors state “our results imply that the KMT2F may be recruited to the rDNA promoter, indirectly through WDR82.” But they should either test this using knockdown of WDR82 or express an RRM-deletion mutant of KMT2F to test this.

---

## [Decision Letter · Decision Letter 2]

2 Apr 2026

Dear Shweta,

Thank you for your continued patience while we considered your revised manuscript "H3K4me3 methyltransferase KMT2F promotes pre-initiation complex formation by RNA Polymerase I to regulate ribosomal RNA transcription" for publication as a Research Article at PLOS Biology. Please accept my sincere apologies for the delays that you have experienced during this round of the peer review process. This revised version of your manuscript has been evaluated by the PLOS Biology editors, the Academic Editor and the original reviewers.

Based on the reviews, I am pleased to say that we re likely to accept this manuscript for publication, provided you satisfactorily address the remaining points raised by Reviewer #2. In addition, I would be grateful if you could please address the following data and other policy-related requests that I have provided below (A-I):

(A) We routinely suggest changes to titles to ensure maximum accessibility for a broad, non-specialist readership. In this case, we would suggest a minor edit to the title, as follows. Please ensure you change both the manuscript file and the online submission system, as they need to match for final acceptance:

“The KMT2F histone methyltransferase interacts with the RNA polymerase I machinery to promote ribosomal RNA transcription”

(B) Thank you for already providing a supplementary file containing the underlying data for the figure panels. However, we note that some data is missing for the supplementary figures or in some cases the tabs are mis-labelled. I would be grateful if you could ensure that the data file provides the underlying data for the following figures:

Figure 1D-E, 2B-C, 4A-D, 4F-H, 5B-F, 6A-D, 7B-D, S3A-E, S4A-D, S6D-E, S6G, S6I, S7E-F, S8A-I, S9A-D

(C) Thank you for providing the ChIP-seq data in the GEO database (GSE261933). However, we note that the data is currently on hold for release. We ask that you please make the data publicly available at this stage before publication.

(D) Please also ensure that each of the relevant figure legends in your manuscript include information on *WHERE THE UNDERLYING DATA CAN BE FOUND*, and ensure your supplemental data file/s has a legend.

(E) We require the original, uncropped and minimally adjusted images supporting all blot and gel results reported in the following Figures:

Figure 3B-E, 4E, 5A, 7A, S1A-D, S6A-C, S7B-D, S8D, S9E

We will require these files before a manuscript can be accepted so please prepare and upload them now. Please carefully read our guidelines for how to prepare and upload this data: https://journals.plos.org/plosbiology/s/figures#loc-blot-and-gel-reporting-requirements.

(F) Per journal policy, if you have generated any custom code during the course of this investigation, please make it available without restrictions. Please ensure that the code is sufficiently well documented and reusable, and that your Data Statement in the Editorial Manager submission system accurately describes where your code can be found. More information on our Code Policy, what and how to share can be found here: https://journals.plos.org/plosbiology/s/code-availability

(G) Please ensure that your Data Statement in the submission system accurately describes where your data can be found and is in final format, as it will be published as written there.

(H) Please ensure that you are using best practice for statistical reporting and data presentation. These are our guidelines https://journals.plos.org/plosbiology/s/best-practices-in-research-reporting#loc-statistical-reporting and a useful resource on data presentation https://journals.plos.org/plosbiology/article?id=10.1371/journal.pbio.1002128

- If you are reporting experiments where n ≤ 5, please plot each individual data point.

(I) Please note that per journal policy, the model system/species studied should be clearly stated in the abstract of your manuscript.

We expect to receive your revised manuscript within three weeks.

*Published Peer Review History*

*Press*

Best wishes,

Richard

Richard Hodge, PhD

rhodge@plos.org

Reviewer remarks:

Reviewer #1: The authors have made substantial efforts to clarify their initial version of this manuscript and to provide compelling additional data and analyses.

Given these additions, I agree with the authors that it would be unreasonable to go even further within the scope of this study.

I thus recommend this manuscript in its current form for publication in PLoS Biology and congratulate the authors for their excellent work.

Reviewer #2: In this manuscript, the authors sought to investigate the role of H3K4me3 at ribosomal DNA (rDNA) during transcription. They identify 1) H3K4me3 appears to be present and potentially enriched at rDNA, 2) KMT2F as the primary methyltransferase of H3K4me3 at rDNA, and 3) loss of KMT2F and H3K4me3 decreases rDNA transcription.

I feel the authors have sufficiently addressed our major concerns, albeit while raising several other smaller issues:

1) In supplemental figure 1, co-localization of NPM1 and KMT2A/F are more convincing than their previous data and consistent with their statement that these factors localize to the nucleolus. While the images appear to show a more substantial decrease in KMT2 levels than their western blots, there is less staining in the siKMT conditions. Intriguingly, given their proposal of the role of KMT2 in the regulation of rDNA transcription, the organization of the nucleolus appears unchanged upon KMT2 knockdown.

2) Another issue we raised was the fact that loss of KMT2 may change transcription of ribosome biogenesis factors. To address this, the authors re-analyzed previously published RNAseq data from HeLa cells with knockdown of KMT2A/F. Of note, HeLa cells were not the model used for many of the other experiments. While it is very surprising to see that the nuclear enrichment scores for myriad ribosome biogenesis GO terms appear to be exactly the same at 0, it does appear to suggest that the majority of pathways regulated by KMT2 are independent of ribosome biogenesis. Of note, it is unclear what the "Significance" color codes are for in Supp. Fig. 5G,H. Also, I would request the authors delineate what statistical cutoffs were to generate these plots.

3) To determine whether KMT2A/F and H3K4me3 are present or enriched at rDNA, the authors re-analyzed ChIPseq data to identify the proportion of these factors or modifications across the genome (Supp Fig 5A-F). While their data showed that KMT2A/F and H3K4me3 are present at rDNA, among other loci, it is slightly difficult to interpret because these bins (intergenic, gene bodies, rDNA, promoter) are not equivalent in the proportion of the genome. For example, gene bodies encompass ~1-2% of the genome, yet ~25% of total H3K4me3 and KMT2 are present in these regions. Thus, it is unclear how to interpret the data, which would be more convincing and interpretable if scaled accordingly.

---

## [Editor Report · Decision Letter 3]

17 Apr 2026

Dear Shweta,

On behalf of my colleagues and the Academic Editor, Marcelo Nollmann, I am pleased to say that we can accept your manuscript for publication, provided you address any remaining formatting and reporting issues. These will be detailed in an email you should receive within 2-3 business days from our colleagues in the journal operations team; no action is required from you until then. Please note that we will not be able to formally accept your manuscript and schedule it for publication until you have completed any requested changes.

PRESS

Best wishes,

Richard

Richard Hodge, PhD

rhodge@plos.org

PLOS
